



# The Relationship between Low-Level Cloud Amount and Its Proxies over the Globe by Cloud Types

Jihoon Shin and Sungsu Park

School of Earth and Environmental Sciences, Seoul National University, Seoul, South Korea

**Correspondence:** Sungsu Park (sungsup@snu.ac.kr)

**Abstract.** We extend upon previous work to examine the relationship between low-level cloud amount (LCA) and various proxies for LCA - estimated low-level cloud fraction (ELF), lower-tropospheric stability (LTS), estimated inversion strength (EIS), and estimated cloud-top entrainment index (ECTEI) - by low-level cloud types (CL) over the globe using individual surface and upper-air observations. Individual CL has its own distinct environmental structure, and therefore our extended

analysis by CL can provide insights into the strength and weakness of various proxies and help to improve them.

Overall, ELF performs better than LTS/EIS in diagnosing the variations in LCA among various CLs, indicating that a previously identified superior performance of ELF to LTS/EIS as a global proxy for LCA comes from its realistic correlations with various CLs rather than with a specific CL. However, ELF as well as LTS/EIS has a problem in diagnosing the decrease in LCA when CL0 (no low-level cloud) is reported and the increase of LCA when CL12 (cumulus) is reported over the

deserts where background stratus does not exist. This incorrect diagnosis of CL0 as a cloudy condition is more clearly seen in the analysis of individual CL frequencies binned by proxy values. If CL0 is excluded, all ELF/LTS/EIS have good inter-CL correlations with the amount-when-present (AWP) of individual CLs. In future, an advanced ELF needs to be formulated to deal with the dissipation of LCA when the inversion base height is lower than the lifting condensation level, to diagnose cumulus updraft fraction as well as the amount of stratiform clouds and detrained cumulus, and to parameterize the scale height

as a function of appropriate environmental variables.

## 1   Introduction

During the last decade, there have been extensive efforts to quantify the impact of low-level clouds on the Earth's climate. How-ever, despite its important role in the global radiation budget and hydrological cycle, various cloud-related feedback processes

are not well represented in most general circulation models (GCMs). Because the climate sensitivities of GCMs are strongly dependent on the representation of cloud processes (e.g., Cess et al. (1990), Stephens (2005), Bony and Dufresne (2005), Andrews et al. (2012), Nam et al. (2012), and Brient and Bony (2012)), the correct understanding and accurate parameterizations of cloud processes are critical for the successful simulation of the Earth's future climate.



Numerous studies have attempted to understand the complex physics and dynamic processes controlling the formation and dissipation of marine stratocumulus clouds (MSC) through observational analysis and modeling (see Wood (2012)). Using large-scale environmental variables, several studies have endeavored to find a simple proxy that can diagnose spatial and temporal variations in MSC. Klein and Hartmann (1993) (KH93 hereafter) showed that a lower tropospheric stability, LTS≡

$\theta_{700} - \theta_{1000}$ where $\theta_{700}$ and $\theta_{1000}$ are the potential temperatures at 700 and 1000 hPa levels, respectively, well correlates with the seasonal variations in LCA in the subtropical marine stratocumulus deck. The observed empirical relationship between LTS and subtropical LCA was used to parameterize LCA in some GCMs (Slingo (1987); Collins et al. (2004)) or evaluate GCMs (Park et al., 2014). Based on the decoupling hypothesis (e.g., Augstein et al. (1974), Albrecht et al. (1979), Betts and Ridgway (1988), Bretherton (1992), and Park et al. (2004)), Wood and Bretherton (2006) (WB06 hereafter) suggested an

estimated inversion strength (EIS) as an alternative proxy for LCA in the subtropical and midlatitude marine stratocumulus decks. Although uncertainty exists regarding whether the observed relationship between EIS and LCA still maintains in future climate, EIS has been used to predict the variations in LCA in response to the climate changes (Caldwell et al. (2013), Qu et al. (2014, 2015)). More recently, Kawai et al. (2017) proposed an estimated cloud-top entrainment index (ECTEI) as a proxy for MSC, which is a modified EIS that takes into account a cloud-top entrainment criteria.

Although the aforementioned proxies (i.e., LTS, EIS, and ECTEI) have been shown to be extremely useful in diagnosing the variations in MSC over the subtropical and midlatitude oceans, their applicability in the other regions (e.g., lands, tropics, and high latitude regions) has been in question. Park and Shin (2019) (PS19 hereafter) found that these proxies are not strongly correlated with the observed LCA when the analysis domain is extended over the entire globe and suggested an estimated low-level cloud fraction (ELF) as a new proxy for the analysis of the spatiotemporal variations in the global LCA. ELF is

defined as ELF=$f \cdot (1 - \sqrt{z_{LCL} \cdot z_{inv}}/\Delta z_s)$, where $f = max[0.15, min(1, q_{v,ML}/0.003)]$ is a freezedry factor with the water vapor specific humidity in the surfaced-based mixed layer, $q_{v,ML}$ in [g kg$^{-1}$]; $z_{LCL}$ is the lifting condensation level (LCL) of near-surface air; $z_{inv}$ is the inversion height estimated from the decoupling hypothesis suggested by Park et al. (2004); and $\Delta z_s = 2750$ [m] is a constant scale height. PS19 showed that ELF is superior to LTS, EIS, and ECTEI in diagnosing the spatial and temporal variations in the seasonal LCA over both the ocean and land, including the marine stratocumulus deck,

and explains approximately 60% of the spatial-seasonal-interannual variance of the seasonal LCA over the globe, which is a much larger percentage than those explained by LTS (2%) and EIS (4%). PS19 also noted several weaknesses of ELF, such as its tendency to underestimate LCA over the deserts and North Pacific and Atlantic oceans and overestimate LCA in other regions.

In this study, we extend PS19 and examine the relationship between LCA and its proxies by individual low-level cloud

types. Individual low-level cloud has its own distinct structure of the planetary boundary layer (PBL) and synoptic environmental conditions (Norris (1998), Norris and Klein (2000)). As the PBL transitions from the well-mixed to a decoupled state, surface-observed low-level clouds change from stratocumulus (CL5 where CL is a low-level cloud code used by surface observers defined from WMO (1975a); see also Park and Leovy (2004)) to cumulus-under-stratocumulus (CL8) and stratocumulus formed by the spreading out of cumulus (CL4), and eventually to shallow (CL1), moderate (CL2), and precipitating deep

cumulus (CL3) with an anvil (CL9). In the stable PBL, sky-obscuring fog (CL11) or fair weather stratus (CL6) are likely to be


observed when the inversion height is slightly higher than $z_{LCL}$ but low-level cloud cannot be formed (CL0) if the inversion height is lower than $z_{LCL}$. In general, fractional area covered by stratiform clouds is larger than that of convective clouds. It is expected that a detailed analysis of the relationship between LCA and various proxies by individual CLs will provide insights regarding the sources of the strengths and weaknesses of various proxies, which may help to develop a better proxy for LCA.

The structure of this paper is as follows. Section 2 briefly explains the conceptual framework of ELF including the data and analysis methods. Section 3 shows the results of the analysis of climatology and seasonal cycle of various CLs and the relationship between the amount-when-present (AWP), frequency (FQ), and amount (AMT) of individual CL and various proxies. Several ways to develop an advanced ELF in future is also discussed. A summary and conclusion are provided in Section 4.

## 2 Method

### 2.1 Conceptual Framework

PS19 provided a detailed description of the definition and physical meaning of various proxies for LCA, which are briefly summarized here. The lower-tropospheric stability (LTS) and estimated inversion strength (EIS) are defined as

$$LTS \equiv \theta_{700} - \theta_{sfc}, \tag{1}$$

$$EIS = LTS + \Gamma_{LCL}^m \cdot z_{LCL} - \Gamma_{700}^m \cdot z_{700}, \tag{2}$$

where $\theta_{700}$ and $\theta_{sfc}$ are the potential temperatures at 700 [hPa] and surface, respectively, and $\Gamma_{LCL}^m$ and $\Gamma_{700}^m$ are the moist adiabatic lapse rates of $\theta$ (in unit of $[K \cdot m^{-1}]$) at the lifting condensation level of near surface air ($z_{LCL}$) and 700 [hPa] height ($z_{700}$), respectively.

The estimated low-level cloud fraction (ELF) is defined as

$$ELF \equiv f \cdot \left[ 1 - \frac{\sqrt{z_{inv} \cdot z_{LCL}}}{\Delta z_s} \right], \tag{3}$$

where $f$ is the freezedry factor (Vavrus and Waliser, 2008) defined as a function of water vapor specific humidity at surface ($q_{v,sfc}$ in unit of $[g \cdot kg^{-1}]$),

$$f = max \left[ 0.15, \ min \left( 1, \frac{q_{v,sfc}}{0.003} \right) \right], \tag{4}$$

and $z_{inv}$ is the inversion height,

$$z_{inv} = - \left( LTS / \Gamma_{700}^m \right) + z_{700} \qquad + \Delta z_s \cdot \left( \frac{\Gamma_{LCL}^m}{\Gamma_{700}^m} \right)$$

$$= - \left( EIS / \Gamma_{700}^m \right) + z_{LCL} \cdot \left( \frac{\Gamma_{LCL}^m}{\Gamma_{700}^m} \right) + \Delta z_s \cdot \left( \frac{\Gamma_{LCL}^m}{\Gamma_{700}^m} \right), \tag{5}$$





where $\Delta z_s = 2750\,[m]$ is a constant scale height. Using the decoupling hypothesis of PLR04, PS19 estimated $z_{inv}$ by assuming that the decoupling parameter $\alpha$ can be parameterized as a linear function of the decoupled layer thickness, $\Delta z_{DL} \equiv z_{inv} - z_{LCL}$,

$$\alpha \equiv \frac{\theta_{inv}^- - \theta_{sfc}}{\theta_{inv}^+ - \theta_{sfc}} \approx \left(\frac{\Delta z_{DL}}{\Delta z_s}\right), \quad 0 \leq \alpha \leq 1, \tag{6}$$

where $\theta_{inv}^+ = \theta_{700} - \Gamma_{700}^m \cdot (z_{700} - z_{inv})$ and $\theta_{inv}^- = \theta_{sfc} + \Gamma_{LCL}^m \cdot (z_{inv} - z_{LCL})$ are the potential temperatures just above and below the inversion height (see Fig. 1 of PS19). In deriving ELF, it was assumed that the top of surface-based mixed layer is identical to $z_{LCL}$. The freezedry factor is designed to reduce the parameterized cloud fraction in the extremely cold and dry atmospheric conditions typical of polar and high latitude winters. ELF can be also written as ELF=$f \cdot [\, 1 - (z_{LCL}/\Delta z_s)\sqrt{1 + (z_{inv} - z_{LCL})/z_{LCL}}\,]$, where $f$ denotes the amount of water vapor in the surface air, $z_{LCL}$ represents

the degree of subsaturation of near-surface air, and $(z_{inv} - z_{LCL})/z_{LCL}$ quantifies the degree of thermodynamic decoupling of the inversion base air from the surface air. ELF predicts that LCA increases as the near-surface air becomes more saturated with enough amount of water vapor and as the PBL becomes more vertically coupled, which is consistent with what is expected to happen in nature. To ensure $0 \leq \alpha \leq 1$ (i.e., thermodynamic scalars at the inversion base ($\theta_{inv}^-$) are bounded by the surface ($\theta_{sfc}$) and inversion top ($\theta_{inv}^+$) properties), the inversion height computed from Eq.(5) was forced to satisfy

$z_{LCL} \leq z_{inv} \leq z_{LCL} + \Delta z_s$.

## 2.2   Data and Analysis

The data used in our study are identical to that used in PS19. The surface observation data are from the Extended Edited Cloud Report Archive (EECRA, Hahn and Warren (1999)), which compiles individual ship and land observations of clouds, present weather, and other coincident surface meteorologies every 3 or 6 hours. The upper-level meteorologies (e.g., $p$ and $\theta$)

are from the ERA interim reanalysis products (ERAI, Simmons et al. (2007)) at 6-hourly time intervals. Spatial and temporal interpolations are performed to compute the upper-level meteorologies at the exact time and location at which the EECRA surface observers reported the LCA. Our analysis uses the data from January 1979 to December 2008 (30 years) over the ocean and January 1979 to December 1996 over land (18 years). Using the 6-hourly ERAI vertical profile of $\theta$ and water vapor interpolated to individual EECRA surface observations, we computed the seven proxies for LCA (i.e., LTS, EIS, ECTEI, ELF,

$\alpha$, $z_{LCL}$, and $z_{inv}$).

    The surface observer reports cloud type (CL) and fractional area (LCA) of low-level clouds following a strict hierarchy from the World Meteorological Organization (WMO (1975b). Table 1). In addition to the ten CL types defined by WMO, EECRA defines two more CL types (CL10, sky-obscuring thunderstorm and shower, and CL11, sky-obscuring fog) by combining the present weather code with the observation of missing CL. Consequently, an individual EECRA observation contains 12 CLs

(from CL0 to CL11) and associated LCA (from 0 to 8 octa), such that a set of 12 CLs forms a complete basis function for the entire EECRA data. Based on similarities in morphology and physical property, we grouped individual CLs into the eight groups, being CL0, CL11, CL6, CL7, CL5, CL84 (Cumulus-with-Stratocumulus), CL12 (Cumulus), and CL39 (Cumulonimbus), in approximately the increasing order of vertical instability. For individual CLs or combinations of CLs, we computed





cloud frequency (FQ), amount-when-present (AWP), and amount (AMT), following the procedures described in Hahn and Warren (1999) and Park and Leovy (2004). Cloud FQ for a specific CL is defined by the fraction of observations reporting the specific CL among the total set of observations reporting any CL information. Cloud AWP is the average LCA when a specific CL is observed. Cloud AMT is the product of FQ and AWP.

Similar to PS19, individual EECRA cloud observations, surface and upper-level meteorologies are averaged into $5^o$latitude x $10^o$longitude seasonal data for each year. To reduce the impact of random noise, a minimum of 10 observations were required to form effective seasonal grid data in each year. These seasonal grid data are used for computing annual climatologies and seasonal differences of various CLs (Fig. 1) and analyzing correlations between the LCA and various proxies by cloud types (Tables 1-2 and Figs. 2-5). In addition, individual EECRA cloud observations are grouped into bins of individual proxies to
better understand the contribution of individual CLs to the overall correlation relationship between the proxies and LCA (Figs. 6-7). ECTEI produced results very similar to those of EIS, such that only the analysis from EIS are shown in this study.

## 3 Results

### 3.1 Climatology and Seasonal Cycle

Figure 1 shows the annual climatology and the differences in the seasonal FQ of various CLs during JJA and DJF. CL0 is
frequently observed over the continents but is rarely reported over the open ocean, implying that the primary factor controlling the formation of low-level clouds is the moisture source at the surface. One of the rare open ocean areas with annual CL0 FQ larger than 10% is the sea surface temperature (SST) cold tongue region in the eastern equatorial Pacific ocean, where SST is lower than the overlying air temperature, net upward buoyancy flux from the sea surface is very weak, and atmospheric PBL is stable (Deser and Wallace, 1990). As a result, turbulent vertical moisture transport from the sea surface to $z_{LCL}$ is strongly
suppressed (i.e., $z_{inv} < z_{LCL}$), resulting in the maximum FQ of CL0 (Park and Leovy, 2004). This indicates that not only the moisture source at the surface, but also vertical stability in the atmospheric PBL controls the formation of low-level clouds. Over the continents and the Arctic area, CL0 is more frequently observed during boreal winters than summers, presumably because strong daytime insolation during summer destabilizes the lower troposphere, promoting the onset of convective clouds (i.e., CL84, CL12, and CL39), strong nocturnal LW radiative cooling during winter stabilizes the lower troposphere, forcing
$z_{inv} < z_{LCL}$, and the amount of moisture at the near surface is very small during winter. Similar to the case over the SST cold tongue, strong vertical stability over the winter continents and Arctic area appears to increase the probability of the occurrence of CL0, which appears to be somewhat opposite to the embedded decoupling processes into ELF that increases as $z_{inv}$ decreases. However, with the freezedry factor, ELF may be able to capture enhanced CL0 frequency over the continents during winter due to a small amount of moisture near the surface.
CL11 is frequently observed over the western North Pacific and Atlantic oceans, including the Arctic area, during JJA when the Arctic sea ice melts and moist warm airs are advected into cold SST region across the midlatitude SST front. This indicates the saturation of advected air parcels by the contact cooling with the underlying cold SST or more upward moisture transport from the open ocean over the Arctic, which can be captured by ELF through the decrease in $z_{LCL}$. CL6 has a





similar climatology and seasonal cycles as CL11, implying that the physical processes controlling the formation of CL11 are similar to those of CL6. CL7 has an annual climatology similar to that of CL6 but its seasonal cycle over the North Pacific and Atlantic oceans is opposite to that of CL6, with more frequency during boreal winters. Similar to CL7, CL39 is more frequently observed during winter in this region, which is presumably due to the frequent passage of midlatitude synoptic

storms in winter. A composite analysis showed that CL39 is frequently observed on the rear side of the midlatitude synoptic cold front with a reduced lower tropospheric stability, while CL7 is observed on the front or near center of synoptic storm with an enhanced lower tropospheric stability (not shown). When the midlatitude storm track passes, anomalous mean vertical motion in the mid-troposphere drives the changes in the mid-level clouds, but the variations in the lower tropospheric stability also drive the changes of LCA, which can be captured by ELF through the variations in $z_{inv}$.

CL5 is more frequently observed over the eastern subtropical and midlatitude oceans during JJA, when the subtropical and midlatitude high is strong and the PBL is relatively well coupled. Over most ocean areas, seasonal variations in CL5 tend to be opposite to those of CL12 and CL39. ELF is designed to capture these conversions between CL5 and CL12 in association with the PBL decoupling. Over northern Asia and Canada, including a portion of the Arctic area, both convective and stratiform clouds are more frequently observed during boreal summers than winters, presumably due to the destabilization of the lower

troposphere by strong insolation heating and more surface moisture.

## 3.2   Proxy vs the AWP of Individual Low-Level Clouds

Figures 2 and 3 show the composite anomalies of LCA and various proxies with respect to the seasonal climatology when a specific CL was reported. The anomalous LCA in the top row ($\Delta$AWP) is the difference between the amount-when-present (AWP) when a specific CL was reported, and climatological LCA. To examine the coherency between $\Delta$AWP and the anoma-

lies of individual proxies in each grid box, we computed the non-centered geographical correlation coefficients between $\Delta$AWP and $\Delta$Proxy over the entire globe (G), ocean (O) and land (L), respectively, which are shown at the top of the individual plots.

When CL0 is reported, AWP is zero, that is, $\Delta$AWP = $-$LCA in Fig. 2a. However, both LTS and EIS increase (G=-0.71 and -0.62 for LTS and EIS, respectively), particularly over the far northern continents and Arctic area. Conversely, ELF decreases in a desirable way, due to the freezedry factor (compare Fig. 2y with 2$\gamma$). Over the eastern subtropical marine stratocumulus

deck, all LTS/EIS/ELF show a hint of negative anomaly which, however, is too weak to explain the substantial decrease in LCA when CL0 is reported. Over the midlatitude oceans, the situation is worse and none of the factors comprising ELF (i.e., $z_{LCL}$, $z_{inv}$, and $\alpha$) can explain the decrease in LCA. Although slightly better than LTS and EIS, ELF has an apparent problem in diagnosing the decrease of LCA when CL0 was reported, particularly over the ocean (O=0.15). This problem worsens without the freezedry factor (Fig. 2y). When CL11, CL6, or CL7 are reported, LCA increases over the entire globe, which are very

well captured by ELF (G=0.97, 0.89, and 0.88 for CL11, CL6, and CL7, respectively), due to the simultaneous decreases in $z_{LCL}$, $z_{inv}$, and $\alpha$. Although slightly worse than ELF, LTS and EIS also captures the increase of LCA when CL11 was reported (G=0.85 and 0.44 for LTS and EIS, respectively). However, undesirable negative anomalies of LTS and EIS over the far northern continents including Arctic area get worse from CL11 to CL6 and CL7, resulting in very weak (G=0.17 for LTS)





or even negative (G=-0.43 for EIS) correlations between ΔLTS/ΔEIS and ΔAWP when CL7 was reported. Overall, ELF is better than LTS and EIS in diagnosing the variations of fog and stratus over both the ocean and land.

In addition to the fog and stratus, ELF captures the variations in LCA in association with CL5 (G=0.74), CL84 (G=0.52), CL12 (G=0.31), and CL39 (G=0.62) reasonably better than LTS and EIS. When CL5 was reported and so LCA increases,
both LTS and EIS increase over the subtropical and midlatitude oceans. However, over the Arctic, Asia, and deserts areas, LTS/EIS shows negative anomalies opposite to the increased LCA, which worsens and extends to other continents from CL5, CL84 to CL12 and CL39, resulting in substantial negative correlations between ΔLTS/ΔEIS and ΔLCA over land for CL84 (L=-0.65/-0.71 for LTS/EIS), CL12 (L=-0.38/-0.38), and CL39 (L=-0.74/-0.80). Although generally better than LTS/EIS, ELF also has a problem in capturing the increase in LCA over Asia and most desert areas when CL12 was reported (L=-0.14). In
summary, an advanced ELF in future should be designed to capture the decrease in maritime LCA associated with CL0 and the increase in continental LCA associated with CL12.

Figure 4 shows the area-averaged seasonal climatology of the AWP and various proxies when a specific CL was reported over the ocean and land during the daytime (9 am - 9 pm) and nighttime (9 pm - 9 am), respectively. By definition, CL11 is always overcast and stratiform clouds tend to have larger AWP than convective clouds. CL39 has larger AWP than CL12,
presumably due to larger cross-sectional/lateral areas of deep convective updraft plumes or the contribution of detrained convective condensates. With the exception of CL39, AWP over the ocean is slightly larger than that over land. The diurnal cycle of the AWP in most CLs is very weak. However, continental CL39 during the night tends to have a slightly larger AWP than during the day, which seems to be contradictory to intuition that deep cumulonimbus over land is forced by strong insolation heating during the day. This may reflect the late evening or nocturnal development of the strongest deep convective system
over the continents in association with the gradual buildup of the mesoscale convective organization forced by the evaporation of convective precipitation (Park (2014a, b)). As a global proxy for the AWP of individual CL, ELF shows more desirable inter-CL variations than LTS and EIS, which have strong ocean-land contrasts (in particular, EIS) and seasonal cycle over land. Due to the freezedry factor, ELF is slightly smaller than $1 - \beta_2$ during DJF over land. ELF has a somewhat stronger diurnal cycle than AWP over land with a larger ELF during the night. The factors comprising ELF ($z_{LCL}$, $z_{inv}$, and $\alpha$) have fairly
similar inter-CL variations with larger values for convective than stratiform clouds. Interestingly, $z_{LCL}$ for CL39 is smaller than that of CL12, presumably due in part to the evaporation of convective precipitation and associated moistening of near surface air when CL39 was reported.

Figure 5 shows the scatter plots of individual CL's AWP as a function of LTS, EIS, $1$-$\beta_2$, and ELF obtained from Fig. 4. If CL0 is excluded, all proxies have very good correlations with the AWP of individual CLs, although ELF and $1 - \beta_2$ perform
slightly better than LTS and EIS. In the case of EIS over land, the regression lines seem to be slightly offset from the data scatters with seemingly too high $R^2$, which is due to CL11 in DJF that has a high EIS located outside of the plotting range. Similar to the regression analysis of PS19, the slope of inter-CL AWP regressed on ELF during the day over the ocean is steeper than that over land. Over the ocean, the regression slopes during the night are roughly similar to those during the day but with systematically higher proxy values. Over land, however, both ELF and $1 - \beta_2$ tend to have steeper regression slopes
during the night than during the day. To be a better proxy for LCA (i.e., LCA=ELF denoted by the dashed grey line), ELF of





CL0 (and CL12 except over land during the day) should be much lower than the current values, while the ELFs of CL5, CL84, CL39 and CL12 over land during the day as well as CL11 and CL67 over the ocean should be higher than the current values. These required behaviors are fairly consistent with the conclusion drawn from the analysis of Figs. 2 and 3.

### 3.3 Proxy vs the FQ of Individual Low-Level Clouds

Figure 6 is the cumulative plot of the frequencies of individual CLs in the bins of various proxies, defined as the number of observations reporting a specific CL type divided by the total observation number in each bin. Figure 6a, a plot with a perfect proxy for LCA, shows that CL0 exists entirely in the zero octa bin, CL11 only exists in the 8 octa bin, and the bin AWP (black line) increases in a perfect linear way as LCA increases, as expected. As LCA increases, the frequency of CL12 decreases but those of stratiform clouds (CL6, CL7, CL5 and CL84) tend to increase. In contrast to CL12, the frequency of CL39 in the low

octa bins gradually increases with LCA. The observation number is relatively large in the zero and 8 octa bins (yellow line). The low-level cloud AMT contributed by individual bin (the cyan line that is a simple product of the black and yellow lines) increases with LCA but not in a perfectly linear way. The overall patterns over land are approximately similar to those over the ocean. Over land, the observation number is the largest in the zero octa bin and convective clouds (CL12 and CL39) are mostly observed during the day. Any good proxy for LCA, if any, should have similar patterns as Figs. 6a and b.

The frequency of CL0 increases as LTS and EIS increase, which is mainly responsible for the undesirable decreases in the AWP and AMT in the high bins of LTS and EIS. Designed as a proxy for marine stratocumulus, however, LTS/EIS reasonably simulates the increase (decrease) in CL5 (CL12) frequency with LTS/EIS over the ocean. In contrast to the case of LCA, CL11 exists in several bins and the frequency of CL39 decreases monotonically with LTS/EIS. Similar to the case of LTS/EIS, CL0 exists ubiquitously in the entire ELF bins, indicating that LTS/EIS/ELF frequently diagnoses the observed no low-level cloud

conditions as cloudy conditions. However, the frequency of CL0 tends to decrease with ELF, such that the bin AWP increases in a desirable way as ELF increases, although the slope is smaller than the case of LCA. The frequency of CL0 in the nonzero ELF bins over land is substantially higher than that over the ocean. The observation number FQs in the zero and 8 octa ELF bins are substantially lower than those in the LCA bins but higher in the intermediate bins, implying that an advanced ELF needs to transfer the observation number FQ in the intermediate ELF bins into the zero octa bin (e.g., by correctly diagnosing

CL0 condition) and 8 octa bin (e.g., by correctly diagnosing CL11 condition).

Table 2 shows spatial-seasonal correlation coefficients between the frequency of individual CL and various proxies. In contrast to Figs. 2 and 3, Table 2 (also Table 3) shows a conventional centered-correlation between the seasonal climatologies (i.e., averaged over all observations) of various proxies and individual CL frequency. LCA increases as the frequencies of sky-obscuring fog (CL11), stratus (CL6, CL7), stratocumulus (CL5, CL84), and continental convective clouds (CL12, CL39)

increase, and decreases as the frequencies of CL0 and marine convective clouds increase. Except for marine CL84 and continental CL12, ELF reproduces these correlation characteristics of LCA with individual CL well, at least qualitatively. The freezedry factor substantially contributes to the improved correlations of CL0 with ELF from $\beta_2$. Over the globe, CL0 is negatively correlated with $z_{inv}$ and $\alpha$ (not shown), presumably due in part to the frequent occurrence of CL0 on the west coast of the major continents and equatorial SST cold regions where $z_{inv}$ is low due to cold SST. Designed as a proxy for marine





stratocumulus, LTS/EIS show a strong correlation with CL5 FQ over the ocean. However, the correlation characteristics of LTS/EIS with other CLs are less desirable than that of ELF. For example, the correlations of LTS/EIS with CL11, CL6, and CL7 over the globe and continental CL5 are significantly weaker than those of LCA and the correlation signs with CL0, CL84, and continental CL12 and CL39 are opposite to those of ELF and LCA. One of the most undesirable aspects of LTS and EIS is a strong positive correlation with CL0 FQ, as was shown in Fig. 6.

### 3.4 Proxy vs the AMT of Individual Low-Level Clouds

Figure 7 is the cumulative plot of the AMT of individual CLs in the bins of various proxies. The bin cloud AMT (the cyan line) increases monotonically with LCA with the largest increase from the 7 to 8 octa-bin (Fig. 7a, b). In the low bins, convective clouds contribute to the cloud AMT more than stratiform clouds but in the high bins, stratiform clouds contribute more. Total cloud AMT (i.e., the integration of the cyan line across the entire bins) over the ocean is larger than that over land. In the 8 octa bin over land, CL39 contributes more than 20% to the cloud AMT. In contrast to LCA, none of the proxies show a required monotonic increase in the bin cloud AMT. Over the ocean, EIS shows an undesirable monotonic decrease in the bin cloud AMT, LTS is slightly better than EIS, and ELF shows a further improvement with the maximum bin cloud AMT shifting to the higher bin. The improvement from EIS/LTS to ELF is more pronounced over land but the rapid decrease in bin cloud AMT from the 7 to 8 octa ELF bins is problematic. These variations in the bin cloud AMT are largely controlled by the variations in the bin cloud FQ (see the yellow line in Fig. 6). All proxies show the increase in the relative contribution of stratiform clouds to the bin cloud AMT as the bin value increases but the contribution of CL39 AMT in the 8 octa bin over land is smaller than that of LCA.

Table 3 shows spatial-seasonal correlation coefficients between the AMT of individual CL and various proxies. The overall correlation characteristics of the cloud AMT are very similar to those of the cloud FQ shown in Table 2. LCA tends to increase as the cloud AMT of individual CL increases. The only exception is marine CL12 AMT that decreases as LCA increases. ELF reproduces these correlation characteristics of the AMT of individual CL with LCA well. As a global proxy for LCA, the correlation characteristics of LTS/EIS with individual cloud AMT are less desirable than that of ELF: the correlations with continental CL12 and CL84 are unrealistically negative and the correlations with sky-obscuring fog and stratus are much weaker than those of ELF and LCA. Table 3 indicates that a superior performance of ELF to LTS/EIS as a global proxy for LCA discovered by PS19 (see the bottom row of Table 3) is derived from its realistic correlations with various CLs rather than with a specific CL.

### 3.5 What are necessary to further improve ELF as a global proxy for LCA ?

We have shown that generally, ELF diagnoses the inter-CL variations in LCA better than LTS/EIS. However, we also identified several weaknesses in ELF, such as the increase in ELF over the ocean when CL0 was reported, and the decrease in ELF over the deserts and Asian continents when CL12 was reported and so LCA increases. In this section, we examine in more details why ELF shows undesirable correlations with LCA for some cases and then provide a potential pathway to further improve ELF in future.





When CL0 is reported, ELF increases over the North Pacific and North Atlantic oceans, which results in a very weak non-centered correlation over the ocean (O=0.15) between $\Delta$LCA (Fig. 2a) and $\Delta$ELF (Fig. 2$\gamma$). Although the correlation over land (L=0.65) is higher than over the ocean, the magnitude of $\Delta$ELF is much smaller than $\Delta$LCA. As shown in Figs. 5g and 5h, CL0 is the most distinct outlier from the desirable AWP=ELF line (dashed lines) in the inter-CL scatter plots. This mis-diagnosis

of CL0 condition with non-zero ELF is also shown in Figs. 6i and 6j and it worsens over land during the night. To understand this problem, we plotted the probability density function (PDF) of $z_{DL} \equiv z_{inv} - z_{LCL}$ using individual observations reporting CL0 and compared it with the PDF of entire observations (CLM) over the ocean (Fig. 8a) and land (Fig. 8b), respectively. As shown, the PDF of near zero $z_{DL}$ when CL0 was reported is higher than that of CLM and the difference over land is larger than that over the ocean. Conceptually, if $z_{DL} < 0$ and so $z_{inv} < z_{LCL}$, low-level cloud cannot be formed, such that LCA is likely

to be small. However, since our ELF= $f \cdot (1 - \sqrt{z_{inv} \cdot z_{LCL}}/\Delta z_s) = f \cdot [\, 1 - (z_{LCL}/\Delta z_s)\sqrt{1 + z_{DL}/z_{LCL}}\,]$ is formulated as a function of $z_{inv} = max(z_{inv}^*, z_{LCL})$ instead of $z_{inv}^*$ (where $z_{inv}^*$ is the inversion height directly obtained from Eq.(5) without any clipping, such that $z_{inv}^*$ can be lower than $z_{LCL}$), this case of $z_{inv}^* < z_{LCL}$ is diagnosed as a highly cloudy condition in the current ELF. It seems that an advanced ELF needs to be able to simulate the decrease in LCA with the increase in the absolute value of $z_{DL}^* \equiv z_{inv}^* - z_{LCL}$, such as ELF= $f \cdot [\, 1 - (z_{LCL}/\Delta z_s)\sqrt{1 + a \cdot \delta_*^2}\,]$, where $\delta_* \equiv z_{DL}^*/z_{LCL}$ is a generalized

decoupling parameter and $a$ is a positive constant. This approach is likely to relocate the observation frequency of CL0 in the high ELF bins into the low ELF bins (Figs. 6i and j), reduce the large ELF values for CL0 (Figs. 5g and h), and improve the non-centered correlations between $\Delta$ELF and $\Delta$LCA for various CL types including CL0 (Figs. 2 and 3).

Another apparent problem of the current ELF is the decrease in ELF over the desert areas (e.g., Sahara, Australia, and Saudi-Arabia) when CL12 was reported (see Figs. 3c and 3$\epsilon$). In contrast to the ocean where the onset of CL12 is often associated

with the decoupling of PBL and the decreases in overlying marine stratocumulus and LCA (e.g., Bretherton (1992), Park et al. (2004)), the onset of CL12 over the deserts without the background stratocumulus seems to directly increase LCA. In this case, ELF tries to mimic the observed increase in LCA by decreasing LCL (see Fig. 3o) but the larger increases in $z_{inv}$ and associated PBL decoupling seem to offset the impact of the reduced LCL, resulting in the decrease in ELF. Conceptually, current ELF is designed to mainly diagnose the variations in stratiform clouds and detrained cumulus at the inversion base,

not the cumulus updraft plume itself (see Fig. 1 of PS17), which is reflected in part by the higher non-centered correlations between $\Delta$ELF and $\Delta$AWP for stratiform clouds than for convective clouds as shown in Figs. 2 and 3. To further improve the performance of ELF, it seems to be necessary to additionally diagnose the fraction of cumulus updraft plume, particularly, in the regions without background stratiform clouds, such as deserts. Because the onset of CL12 is closely associated with the PBL decoupling, one plausible approach is to incorporate a process to increase ELF as $\delta_*$ increases, such that it can offset

the decreases in stratocumulus and ELF with increasing $\delta_*$. If the aforementioned ELF= $f \cdot [\, 1 - (z_{LCL}/\Delta z_s)\sqrt{1 + a \cdot \delta_*^2}\,]$ is adopted as an advanced ELF, the contribution of cumulus updraft plume can be incorporated by setting $a$ to be smaller (or even negative) than the default case excluding the contribution of cumulus updraft plume. Potentially, $a$ could be parameterized as a decreasing function of $z_{LCL}$.

Figures 8c-f show the variations in $z_{LCL}$, $z_{inv}$, $\sqrt{z_{LCL} \cdot z_{inv}}$, and $\alpha$ as a function of ELF and LCA when CL5 and CL12

were reported over the ocean and land, respectively. When averaged over the entire bins (the 'all' bin in the right most column





in each plot), CL12 has higher $z_{LCL}$, $z_{inv}$, and $\alpha$ than CL5, which is consistent with our conceptual understanding. The increase in CL12 AWP from the zero to one-octa bins over land is accompanied by the rapid increase in $\alpha$ (black solid line in Fig. 8f), presumably reflecting the onset of cumulus updraft plume as the PBL is decoupled which, as mentioned before, is not correctly captured by current ELF (black dotted line in Fig. 8f). For both CL5 and CL12 (and also other CLs, not shown),

$z_{LCL}$ tends to decrease monotonically with LCA and ELF, however, $z_{LCL}$ and $z_{inv}$ decrease more rapidly with ELF than with LCA. As a result, the decreasing rate of $\sqrt{z_{inv} \cdot z_{LCL}}$ with ELF is much larger than that with AWP (green lines in Figs. 8c-f). One simple way to remedy this problem is to parameterize the scale height $\Delta z_s$ in ELF= $f \cdot (1 - \sqrt{z_{inv} \cdot z_{LCL}}/\Delta z_s)$ as a function of appropriate environmental variables, such as $z_{inv}$, $z_{LCL}$, and $q_{v,sfc}$. To check whether this is a possible approach, we computed an ideal scale height $\Delta z_{s,i}$ in an adhoc manner, such that it exactly reproduces the observed LCA.

More specifically, for individual data points shown in Figs. 5g and 5h, we computed $\Delta z_{s,i} = (\sqrt{z_{inv} \cdot z_{LCL}})/(1 - AWP/f)$ by inverting Eq.(3) (here, we implicitly assumed that $\Delta z_s$ used in Eq.(3) for deriving ELF differs from $\Delta z_s = 2750$ [m] used in Eq.(5) for deriving $z_{inv}$, which is a completely reasonable assumption because there is no physical reason for $\Delta z_s$ in both equations to be identical). Figures 8g and 8h show the distribution of $\Delta z_{s,i}$ in the phase space of $z_{LCL}$ and $\delta \equiv z_{DL}/z_{LCL}$ over the ocean and land, respectively. As shown, $\Delta z_{s,i}$ has a large inter-CL spread (and also relatively smaller seasonal and

diurnal spreads) instead of being a constant 2750 [m]. There is a tendency for fog and stratus to have larger $\Delta z_{s,i}$ than CL0 and convective clouds and to the first order, $\Delta z_{s,i}$ seems to increase as $\delta$ increases and $z_{LCL}$ decreases. Various CLs, each of which have their own distinct PBL structure and AWP, seem to be reasonably separated from each other in this phase diagram, implying a possibility to parameterize $\Delta z_s$ as a function of $z_{LCL}$ and $\delta$. Because an advanced ELF needs to incorporate other aspects discussed in the above two paragraphs, which will presumably involve some changes in the functional form of ELF,

we leave a detailed parameterization of $\Delta z_s$ for future research.

## 4  Summary and Conclusion

We extended the previous work of Park and Shin (2019), to examine the relationship between various proxies (i.e., LTS, EIS, ECTEI, and ELF) and LCA of individual low-level cloud types (CL). An individual CL has its own distinct PBL structure, such that detailed analysis of the relationship between various proxies and LCA of individual CL can provide insights into the

strength and weakness of individual proxies, which may help to develop a better proxy in future.

Firstly, we compared the annual climatology and seasonal cycle of individual CL's frequency (Fig. 1). CL0 is frequently reported over the winter continents and Arctic area but is seldom reported over the open ocean except in the eastern equatorial SST cold tongue region where PBL is stable in association with negative surface buoyancy flux. By construction, ELF has a limitation in correctly diagnosing reduced cloudiness with enhanced stability in this region. CL11 and CL6 are frequently

observed over the summer western North Pacific/Atlantic oceans and Arctic area, presumably due in part to the cooling of northward advected air parcels and enhanced upward moisture flux through the ice-free Arctic ocean during summer. These processes can be captured by ELF through the decrease in $z_{LCL}$. Over the North Pacific and Atlantic oceans, CL7 and CL39 are more frequently observed during DJF in association with the frequent passage of synoptic storms and the formation of CL7





(CL39) on the front (rear) side of warm (cold) front where lower tropospheric stability is higher (smaller) than the climatology, which can be captured by ELF through the changes of $z_{inv}$. CL5 is frequently observed over the eastern subtropical and midlatitude oceans during JJA and inter-seasonal variations in CL12 and CL39 over most ocean areas tend to be opposite to those of CL5. ELF is designed to capture these conversions between stratocumulus and cumulus in association with the PBL
decoupling.

We then examined the relationship between the anomalies of various proxies and AWP with respect to the climatology when a specific CL was reported in each grid box (Figs. 2 and 3). When CL0 was reported, LTS/EIS does not capture the decrease in LCA and ELF has a similar problem except over the northern continents during winter where the freezedry factor operates. When stratiform clouds are reported, ELF captures the increase in LCA very well due to the simultaneous decreases
in $z_{LCL}$, $z_{inv}$, and $\alpha$. With the exception of over the far northern continent and Arctic area, LTS/EIS works well also, but their performance for CL6 and CL7 are degraded mainly due to undesirable anomalies over the Asia and Arctic area. As well as fog and stratus, ELF captures the variations in LCA when stratocumulus and cumulus are reported reasonably well and significantly better than LTS and EIS. However, when CL12 was reported over Asia and most desert areas, ELF, as well as LTS/EIS, had a problem in capturing the increase in LCA. ELF shows more consistent inter-CL variations with the AWP of
individual CL than LTS and EIS, which have too strong ocean-land contrasts and seasonal cycle over land (Fig. 4). The scatter plots between various proxies and individual CL's AWP showed that if CL0 is excluded, all LTS/EIS/ELF have very good correlations with the AWP of individual CLs, although ELF perform slightly better than LTS and EIS (Fig. 5). To be a better proxy for LCA, the ELF for CL0 and CL12 over ocean and nocturnal land should be reduced, while the ELF for CL11 and CL12 over land during the day time should be enhanced.

We also analyzed individual CL's frequency in the bins of various proxies. In the case of the perfect proxy for LCA (i.e., LCA itself), the frequency of CL12 (stratiform clouds) decreases (increases) with LCA; convective clouds are mostly observed during the day, particularly over land; CL0 exists entirely in the zero octa bin; the bin AWP increases in a perfect linear way as LCA increases; and the observation number FQ is the largest in the zero (particularly, over land) and 8 octa bins. Similar to the perfect proxy, all LTS/EIS/ELF simulate the decrease in CL12 FQ (stratiform clouds FQ) from the low to
high bins reasonably. However, all proxies incorrectly diagnose the observed no low-level cloud conditions (CL0) as cloudy conditions (more severely for LTS/EIS), resulting in unrealistic distributions of the bin AWP and observation number FQ across the bins. The analysis of spatial-seasonal correlation reveals that LCA increases as the frequencies of sky-obscuring fog, stratus, stratocumulus, and continental convective clouds increase, and decreases as the frequencies of CL0 and marine convective clouds increase. Except for marine CL84 and continental CL12, ELF reproduces these observed characteristics
much better than LTS/EIS, which, in particular, suffers from an unrealistically strong positive spatial-seasonal correlation with the CL0 frequency. Similar to the aforementioned analysis of CL's frequencies, all LTS/EIS/ELF do not correctly reproduce the observed monotonic increase in the bin cloud AMT, due mainly to the incorrect diagnosis of CL0 as cloudy conditions, although ELF performs better than LTS/EIS. The analysis of spatial-seasonal correlations between the AMT of individual CL and various proxies indicates that a superior performance of ELF to LTS/EIS as a global proxy for LCA comes from its realistic
correlations with various CLs rather than with a specific CL.





Finally, to provide a potential pathway for an advanced ELF in future, we examined in more detail the cases when ELF performs poorly. When CL0 is reported and so LCA decreases, ELF increases undesirably from its climatological value at each grid point, which is speculated to be associated with the constraint that forces $z_{inv}$ to be larger than $z_{LCL}$. Because low-level cloud cannot be formed when the inversion height is lower than $z_{LCL}$, current ELF is likely to mis-diagnose CL0 as

cloudy conditions. It is necessary to allow $z_{DL} = z_{inv} - z_{LCL}$ to be negative and reformulate ELF to appropriate handle the negative $z_{DL}$. When CL12 is reported over the deserts where background stratiform clouds do not exist, LCA increases but ELF decreases undesirably from its climatological value. This is presumably because current ELF is designed to handle the variations in stratiform clouds and detrained cumulus at the inversion base, not the cumulus updraft plume itself. An advanced ELF needs to diagnose the fraction of cumulus updraft plume, also. Current ELF=$f \cdot (1 - \sqrt{z_{inv} \cdot z_{LCL}}/\Delta z_s)$ assumes a

constant scale height, $\Delta z_s$=2750 [m]; however, it turns out that the ideal $\Delta z_s$ allowing ELF to exactly diagnose the observed AWP of individual CLs has a large inter-CL spread, implying a need to parameterize $\Delta z_s$ as a function of appropriate variables, if any. One possible way of addressing these problems is to formulate ELF=$f \cdot [\, 1 - (z_{LCL}/\Delta z_s)\sqrt{1 + a \cdot \delta_*^2}\,]$, where $\delta_* \equiv (z_{inv}^* - z_{LCL})/z_{LCL}$ and $z_{inv}^*$ is allowed to be lower than $z_{LCL}$, and then parameterize $a$ and $\Delta z_s$ as a function of appropriate environmental variables. Although not shown here, we checked that the observed significant correlations between ELF and

LCA were also simulated by the Community Atmosphere Model version 5 (CAM5, Park et al. (2014)) and the Seoul National University Atmosphere Model version 0 with a Unified Convection Scheme [SAM0-UNICON, Park et al. (2019, 2017), Park (2014a, b)], which, in addition to the derivation of an advanced ELF, will be reported in the near future.

*Data availability.* The EERCA cloud data used in our study is available at https://rda.ucar.edu/datasets/ds292.2/

*Author contributions.* Sungsu Park guided the research and Jihoon conducted the overall analysis under the supervision of Sungsu Park.

*Competing interests.* The authors declare that they have no conflict of interest.

*Acknowledgements.* This work was supported by the Creative-Pioneering Researchers Program of Seoul National University (SNU; 3345-20180018).





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

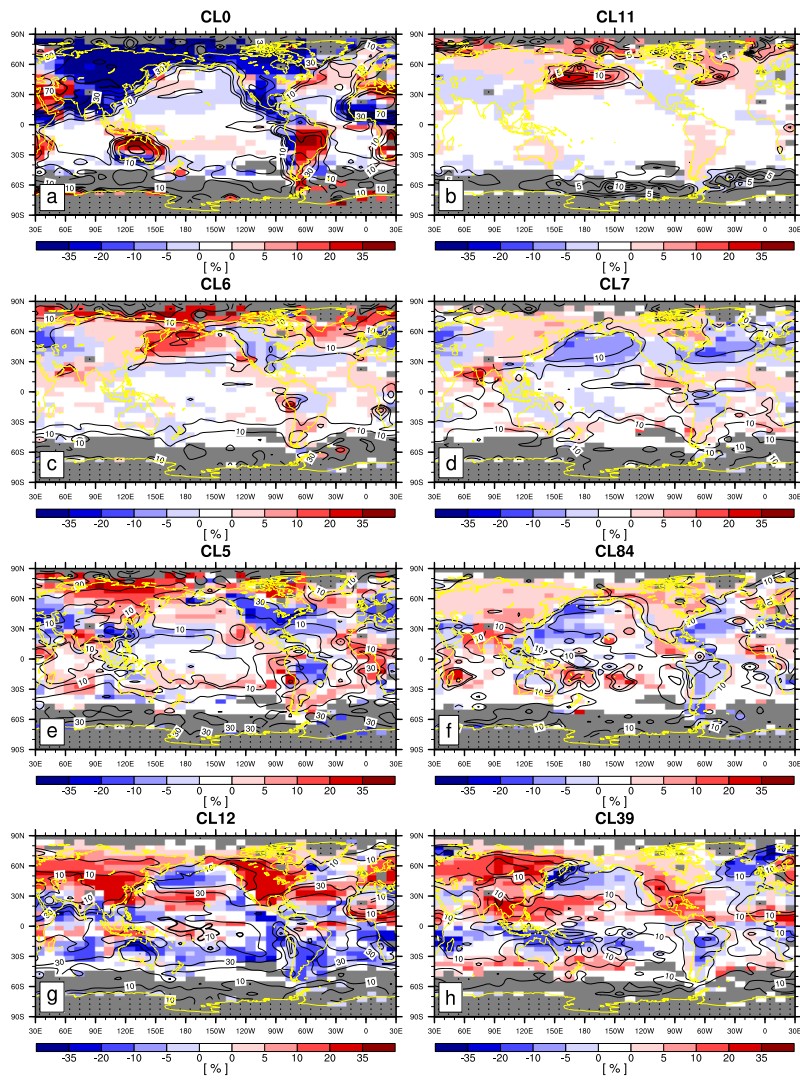

**Figure 1.** (Color) The differences of climatological CL frequency (FQ) between JJA and DJF [ΔFQ = FQ(JJA) - FQ(DJF)] for (a) CL0, (b) CL11, (c) CL6, (d) CL7, (e) CL5, (f) CL84, (g) CL12, and (h) CL39. Solid lines denote annual mean FQ of individual CL with a contour interval of 5%, except Fig. 1b which has an additional contour line of 2.5%. In each plot, statistically insignificant ΔFQ at the 99.9 % confidence level from the two-sided Student t-test assuming independent samples are denoted by white color. Grid boxes with the observation number less than 100 during either JJA or DJF are shaded by gray color. Grid boxes with total observation number less than 100 are marked with a dot.



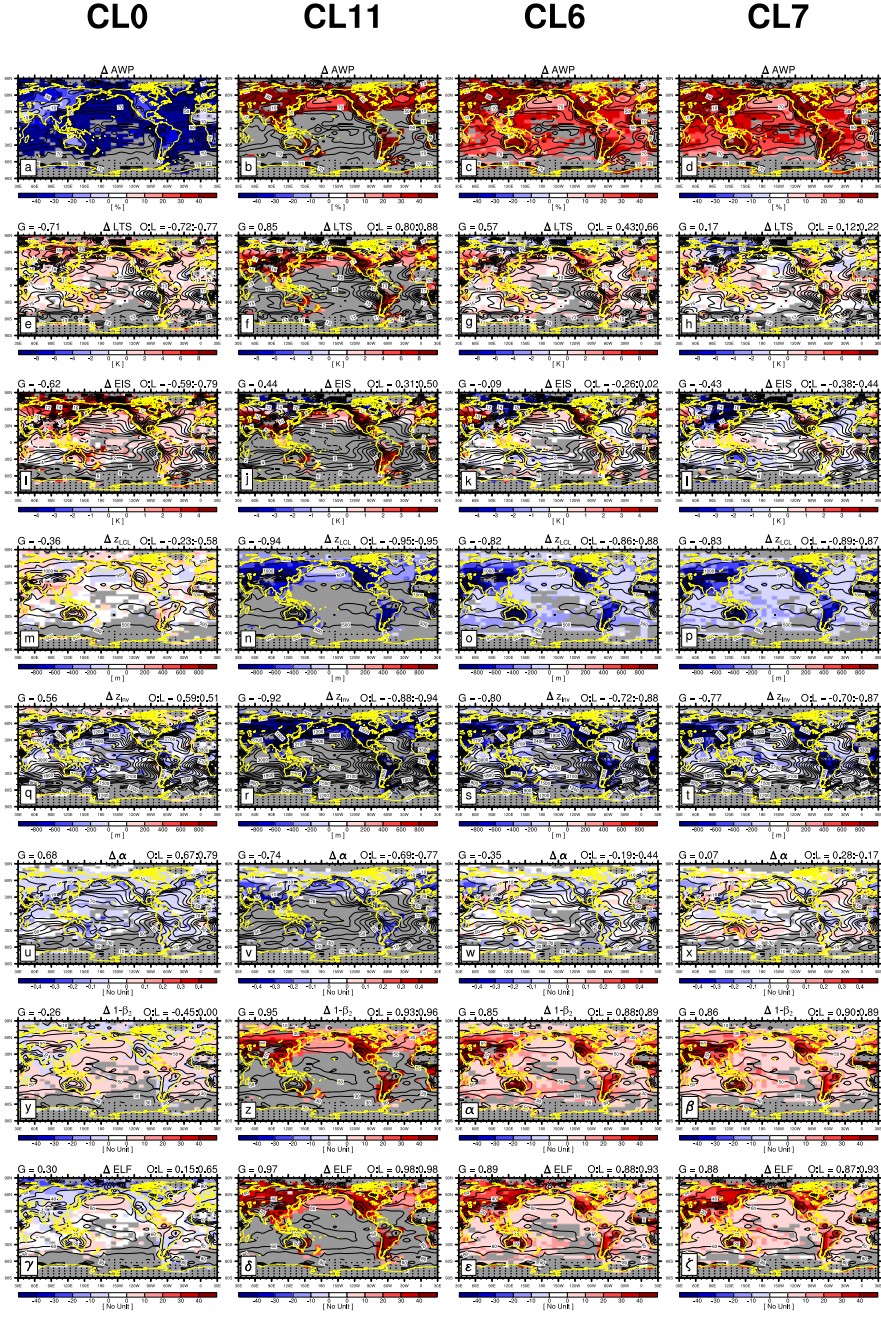

**Figure 2.** Composite anomalies of (1st row) AWP (amount-when-present), (2nd) LTS, (3rd) EIS, (4th) $z_{LCL}$, (5th) $z_{inv}$, (6th) $\alpha$, (7th) $1-\beta_2$, and (8th) ELF with respect to the annual climatology when (first column) CL0, (2nd) CL11, (3rd) CL6, and (4th) CL7 was reported. $\Delta$AWP is the difference between the AWP of a specific CL and climatological LCA. Contour line is the annual climatology of LCA and individual proxies. At the top of individual plot, non-centered correlation coefficients between $\Delta$AWP and $\Delta$proxy over the globe (G), ocean (O) and land (L) are shown. Grid boxes with the observation number of a specific CL less than 100 are shaded by gray color. The other conventions are the same as those of Fig. 1.



**Figure 3.** Same as Fig. 2 but for CL5, CL84, CL12, and CL39.



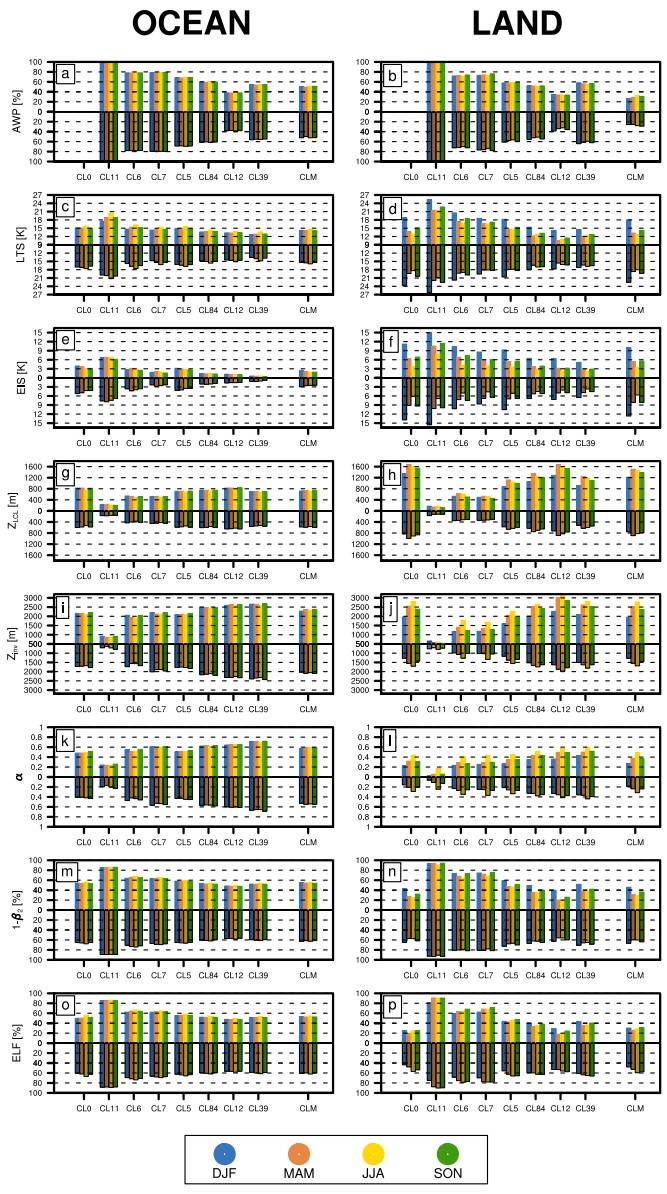

**Figure 4.** Seasonal climatologies of the (top row) AWP and (the other rows) various proxies averaged over the (left) ocean and (right) land for each season (DJF, MAM, JJA, SON denoted by different colors) during the daytime (09 am - 09 pm, upward bars with bright colors) and nighttime (09 pm - 09 am, downward bars with dark colors), respectively, when a specific CL was reported. In each plot, CLM denotes the climatology for all CLs.



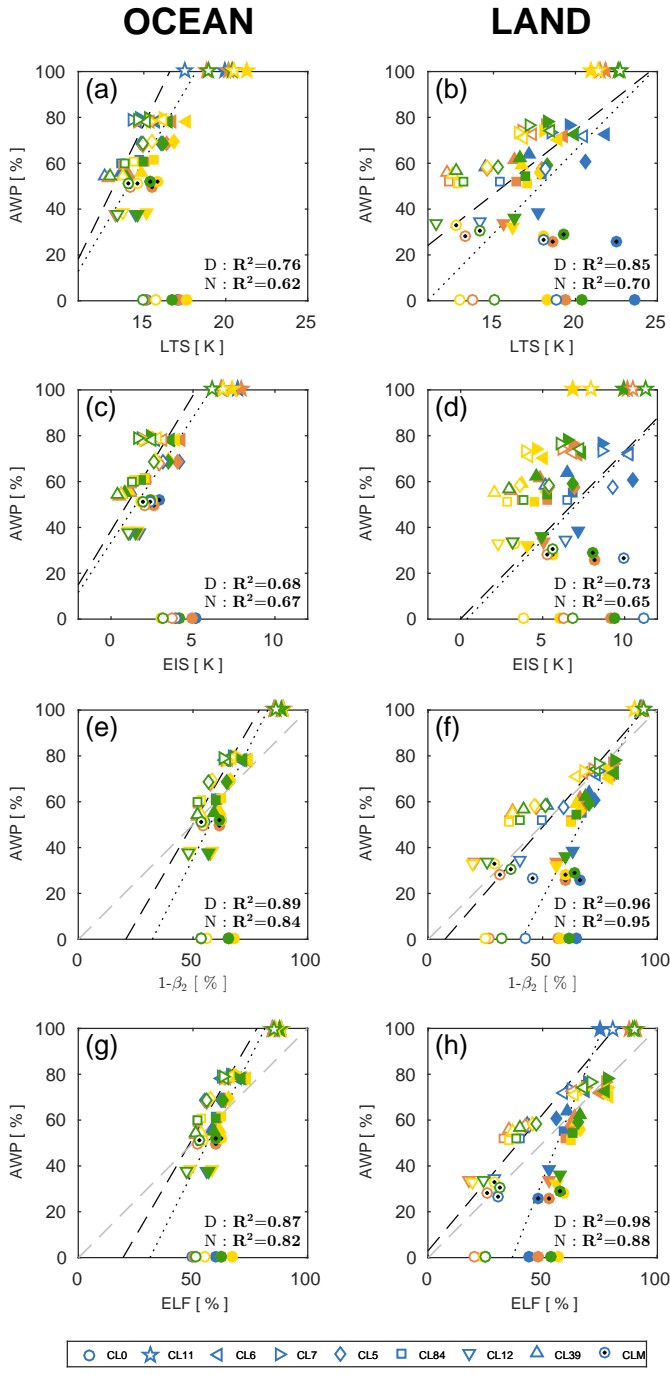

**Figure 5.** Scatter plots of Fig. 4 over the (left) ocean and (right) land, respectively. Also plotted are the linear regression lines and squared correlation coefficients ($R^2$) during the daytime (D, dashed) and nighttime (N, dotted), respectively. The dashed gray lines in the last four plots denote AWP=ELF. The CLM and CL0 cases are not included in the regression analysis.





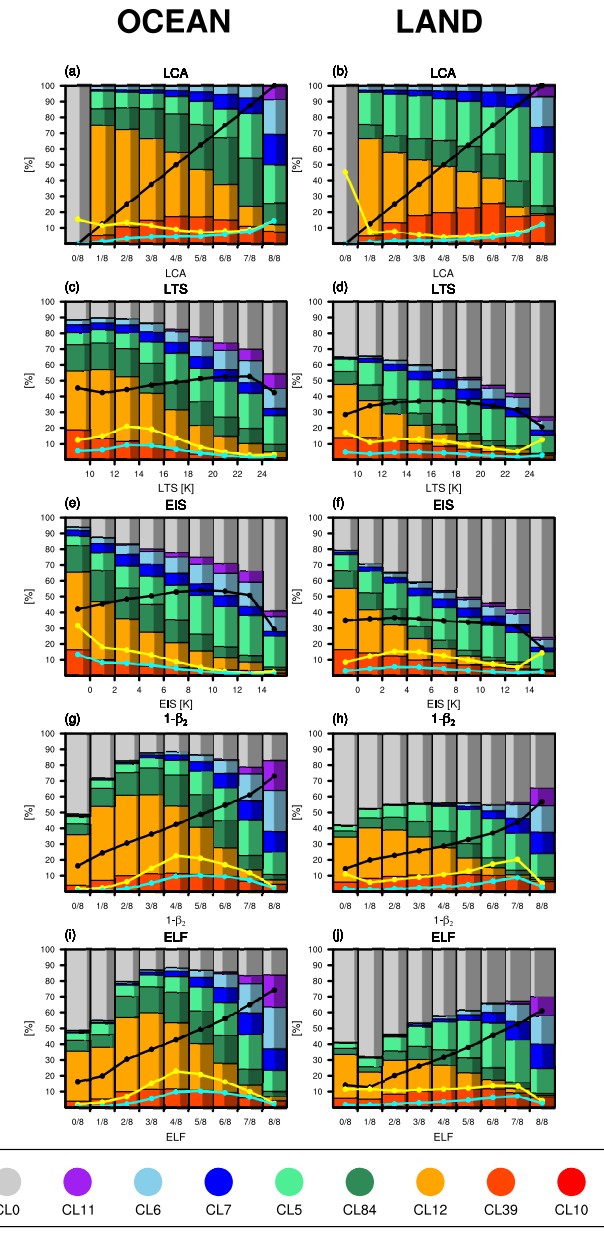

**Figure 6.** Cumulative FQ of individual CLs in the bins of various proxies, (a),(b) LCA (i.e., a perfect proxy for LCA), (c),(d) LTS, (e),(f) EIS, (g),(h) $1-\beta_2$, and (i),(j) ELF over the (left) ocean and (right) land, respectively. AWP of all CLs in each bin is denoted by the black line. The observation number FQ of individual bin (the ratio of the observation number in each bin to the total observation number of entire bins) is denoted by the yellow line. LCA in each bin is denoted by the cyan line, which is the product of the black and yellow lines. The sum of the yellow line integrated over the entire bins is 100. The sum of the cyan line integrated over the entire bins is the global-annual mean LCA. The bright and dark colors in each bar denote the fractions during the daytime and nighttime, respectively.



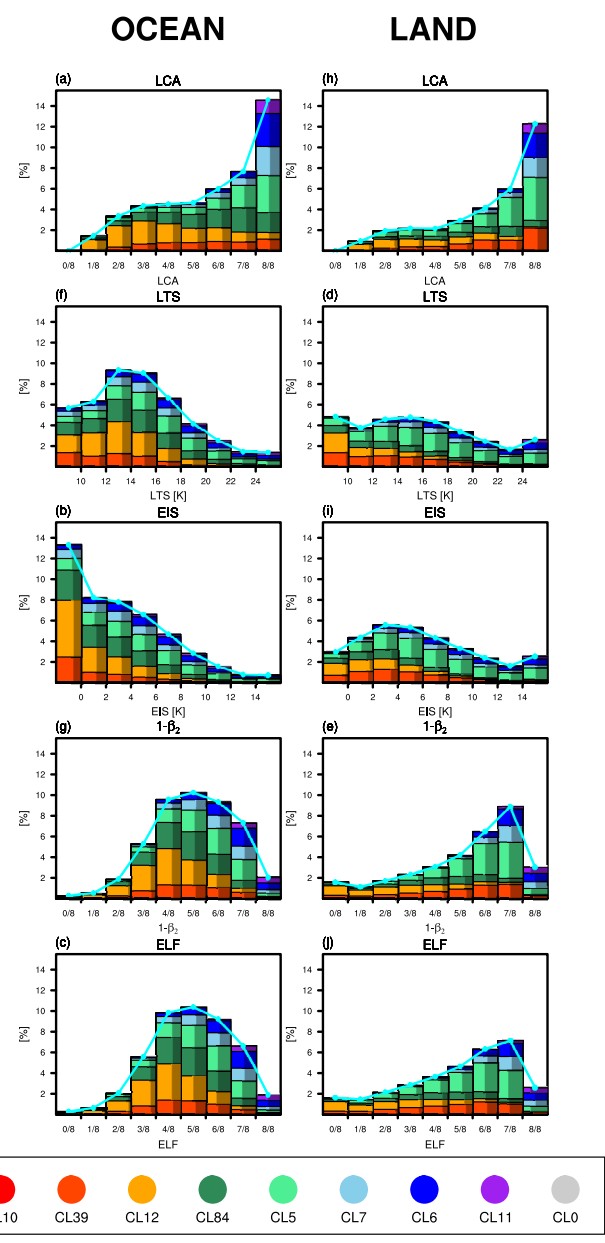

**Figure 7.** Same as Fig. 6 but for cumulative AMT of individual CL in each bin. The cyan lines are identical to those shown in Fig. 6. The sum of all CLs' AMT integrated over the entire bins is the global annual-mean LCA, which is identical regardless of the proxies used for the composite.





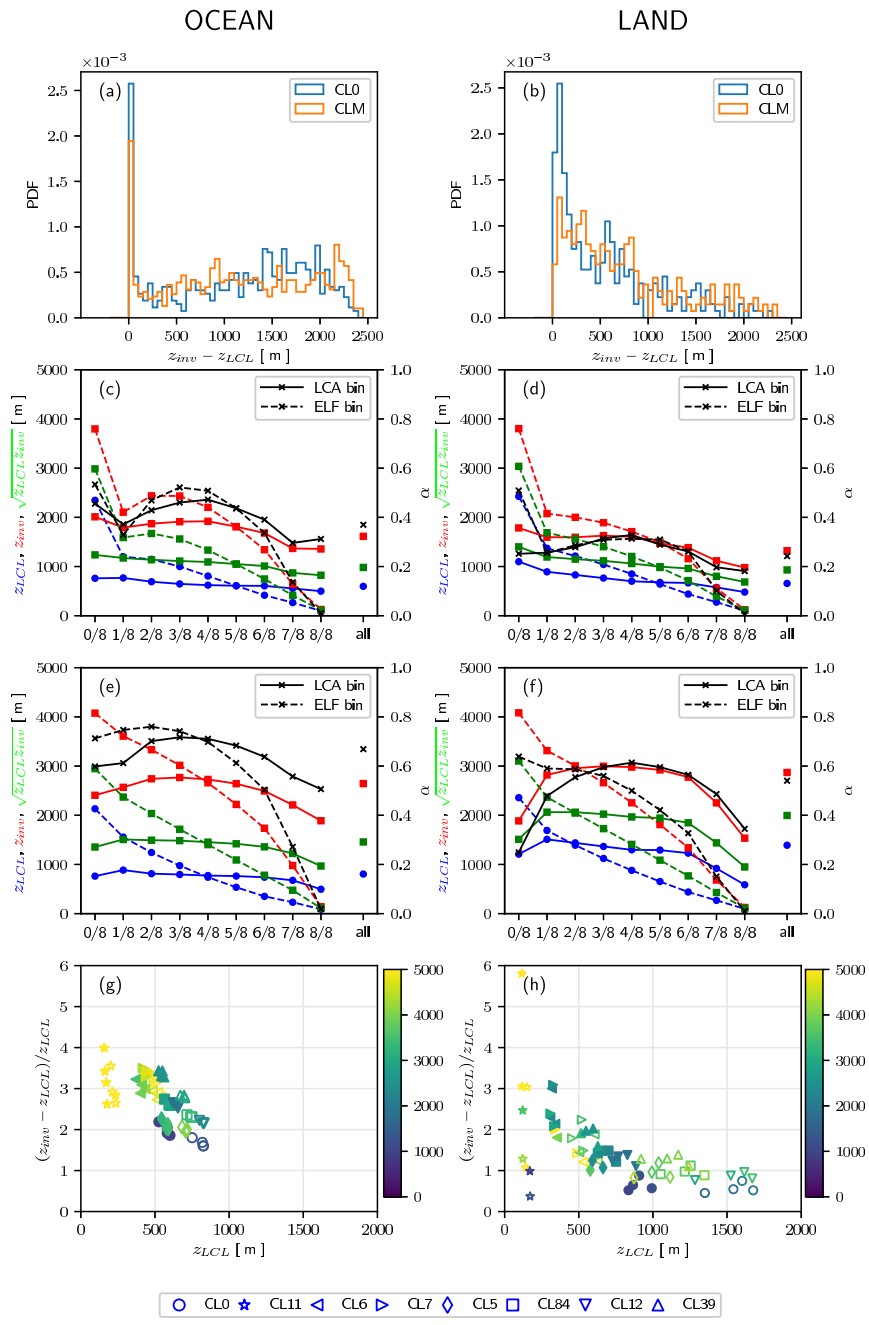

**Figure 8.** (a),(b) Probability density functions (PDF) of $z_{DL} = z_{inv} - z_{LCL}$ when CL0 was reported (blue) and any CL was reported (red); (c)-(f) $z_{LCL}$ (blue), $z_{inv}$ (red), $\alpha$ (black), and $\sqrt{z_{LCL} \cdot z_{inv}}$ (green) in each octa bins of LCA (solid lines) and ELF (dashed lines) when [(c),(d)] CL5 was reported and (e),(f) CL12 was reported, with the values averaged over the entire bins denoted by 'all' in the right most column; and [(g),(h)] the distribution of $\Delta z_{s,i} = (\sqrt{z_{inv} \cdot z_{LCL}})/(1 - AWP/f)$ as a function of $z_{LCL}$ and $\delta \equiv z_{DL}/z_{LCL}$ for individual data points shown in Figs. 5g and 5h. The plots on the left and right columns are over the ocean and land, respectively.



**Table 1.** Low-level cloud (CL) specified by WMO (CL0-CL9). EECRA defined two additional CLs - CL10 and CL11. When multiple CLs exist, the observer is allowed to report only one CL as a representative CL following the coding priority. Among four cloud types (CL1, CL5, CL6, and CL7), the cloud type that has the largest sky fraction has the highest priority. 'Bad weather' denotes the conditions that generally exist during precipitation and a short time before and after.

| CL | Nontechnical Description | Coding Priority | Short Name |
|----|--------------------------|-----------------|------------|
| 0 | No stratocumulus, stratus, cumulus, or cumulonimbus | 10 | No Low-Cloud |
| 1 | Cumulus with little vertical extent and seemingly flattened or ragged cumulus other than of bad weather, or both. | By Cover | Shallow Cumulus |
| 2 | Cumulus of moderate or strong vertical extent, generally with protuberances in the form of domes or towers, either accompanied or not by other cumulus or by stratocumulus, | 5 | Moderate Cumulus |
| 3 | Cumulonimbus, the summits of which at least partially lack sharp outlines but are neither clearly fibrous (cirriform) nor in the form of an anvil; cumulus, stratocumulus, or stratus may also be present | 2 | Cumulonimbus |
| 4 | Stratocumulus formed by the spreading out of cumulus; cumulus may also be present | 3 | Stratocumulus from Cumulus |
| 5 | Stratocumulus not resulting from the spreading out of cumulus | By Cover | Stratocumulus |
| 6 | Stratus in a more or less continuous sheet or layer, or in ragged shreds, or both, but no stratus fractus of bad weather | By Cover | Fair Weather Stratus |
| 7 | Stratus fractus of bad weather or cumulus fractus of bad weather, or both (pannus), usually below altostratus or nimbostratus | By Cover | Bad Weather Fractus |
| 8 | Cumulus and stratocumulus other than that formed from the spreading out of cumulus; the base of the cumulus is at a different level from that of the stratocumulus | 4 | Cumulus under Stratocumulus |
| 9 | Cumulonimbus, the upper part of which is clearly fibrous (cirriform) often in the form of an anvil, either accompanied or not by cumulonimbus without anvil or fibrous upper part, by cumulus, stratocumulus, stratus, or pannus | 1 | Cumulonimbus with Anvil |
| 10 | Sky is obscured (CL=missing with total cloud fraction N=9) by thunderstorm shower (ww=80-99) | · | Sky-obscuring TS (Thunderstorm Shower) |
| 11 | Sky is obscured (CL=missing with total cloud fraction N=9) by fog (ww=10-12, 40-49) | · | Sky-obscuring Fog |





**Table 2.** Spatial-seasonal correlation coefficients between various proxies and the frequency (FQ) of individual CL. In contrast to Figs. 2 and 3 where non-centered correlation coefficients were computed, the values in this table are the conventional centered-correlation coefficients computed from the climatological seasonal proxies obtained by using all observations in each seasonal grid box instead of the observations reporting a specific CL. In this table, LCA is a perfect proxy for LCA. Statistically significant correlations at the 99.9 % confidence level from the Student $t$ test assuming independent samples are denoted by the bold characters.

| CL | Domain | LTS | EIS | $1 - \beta_2$ | ELF | LCA |
|------|--------|-------|-------|-------|-------|-------|
| CL0 | O | **0.69** | **0.79** | **0.42** | **-0.46** | **-0.62** |
| | L | **0.28** | **0.47** | **-0.33** | **-0.69** | **-0.87** |
| | G | **0.46** | **0.64** | **-0.19** | **-0.67** | **-0.82** |
| CL11 | O | **0.45** | **0.23** | **0.55** | **0.63** | **0.49** |
| | L | **0.22** | **0.15** | **0.41** | **0.41** | **0.37** |
| | G | **0.20** | **0.07** | **0.47** | **0.55** | **0.53** |
| CL6 | O | **0.32** | **0.52** | **0.75** | **0.70** | **0.56** |
| | L | **0.27** | **0.14** | **0.46** | **0.47** | **0.45** |
| | G | **0.22** | **0.27** | **0.61** | **0.60** | **0.54** |
| CL7 | O | **-0.15** | 0.15 | **0.36** | **0.47** | **0.70** |
| | L | 0.01 | -0.00 | **0.43** | **0.52** | **0.56** |
| | G | **-0.16** | **-0.06** | **0.38** | **0.52** | **0.69** |
| CL5 | O | **0.40** | **0.59** | **0.66** | **0.39** | **0.31** |
| | L | **0.17** | **0.15** | **0.57** | **0.54** | **0.68** |
| | G | **0.30** | **0.40** | **0.56** | **0.36** | **0.31** |
| CL84 | O | 0.01 | **-0.12** | **-0.08** | 0.03 | **0.28** |
| | L | **-0.29** | **-0.50** | **-0.07** | **0.18** | **0.33** |
| | G | **-0.22** | **-0.43** | 0.05 | **0.27** | **0.50** |
| CL12 | O | **-0.36** | **-0.79** | **-0.78** | **-0.67** | **-0.53** |
| | L | **-0.49** | **-0.68** | **-0.30** | 0.01 | **0.19** |
| | G | **-0.45** | **-0.75** | **-0.30** | -0.03 | **0.10** |
| CL39 | O | **-0.46** | **-0.38** | **-0.20** | **-0.21** | -0.08 |
| | L | **-0.17** | **-0.17** | **0.14** | **0.21** | **0.35** |
| | G | **-0.32** | **-0.31** | 0.03 | 0.08 | **0.17** |
| CLM | O | - | - | - | - | - |
| | L | - | - | - | - | - |
| | G | - | - | - | - | - |



**Table 3.** Same as Table 2 but for the amount (AMT) of individual CL.

| CL | Domain | LTS | EIS | $1 - \beta_2$ | ELF | LCA |
|----|--------|-----|-----|---------------|-----|-----|
| CL0 | O | - | - | - | - | - |
| | L | - | - | - | - | - |
| | G | - | - | - | - | - |
| CL11 | O | **0.45** | **0.23** | **0.55** | **0.63** | **0.49** |
| | L | **0.22** | **0.15** | **0.41** | **0.41** | **0.37** |
| | G | **0.20** | **0.07** | **0.47** | **0.55** | **0.53** |
| CL6 | O | **0.32** | **0.51** | **0.76** | **0.72** | **0.60** |
| | L | **0.29** | **0.18** | **0.48** | **0.49** | **0.48** |
| | G | **0.22** | **0.27** | **0.62** | **0.62** | **0.58** |
| CL7 | O | **-0.14** | **0.17** | **0.39** | **0.49** | **0.73** |
| | L | 0.02 | 0.02 | **0.44** | **0.52** | **0.57** |
| | G | **-0.14** | **-0.03** | **0.40** | **0.53** | **0.71** |
| CL5 | O | **0.43** | **0.58** | **0.70** | **0.47** | **0.45** |
| | L | **0.23** | **0.19** | **0.61** | **0.58** | **0.72** |
| | G | **0.33** | **0.39** | **0.62** | **0.46** | **0.46** |
| CL84 | O | **0.09** | -0.00 | **0.07** | **0.18** | **0.47** |
| | L | **-0.24** | **-0.46** | 0.00 | **0.24** | **0.41** |
| | G | **-0.17** | **-0.37** | **0.14** | **0.35** | **0.61** |
| CL12 | O | **-0.34** | **-0.74** | **-0.70** | **-0.59** | **-0.36** |
| | L | **-0.44** | **-0.63** | **-0.21** | **0.07** | **0.28** |
| | G | **-0.43** | **-0.73** | **-0.23** | **0.03** | **0.22** |
| CL39 | O | **-0.37** | **-0.16** | -0.00 | **-0.06** | **0.08** |
| | L | **-0.08** | **-0.04** | **0.26** | **0.28** | **0.40** |
| | G | **-0.22** | **-0.13** | **0.17** | **0.15** | **0.23** |
| CLM | O | **-0.20** | 0.01 | **0.48** | **0.81** | **1.00** |
| | L | **-0.06** | **-0.21** | **0.58** | **0.82** | **1.00** |
| | G | **-0.23** | **-0.23** | **0.54** | **0.84** | **1.00** |