# Peer review of "The Relationship between Low-Level Cloud Amount and Its Proxies over the Globe by Cloud Types"

_Atmospheric Chemistry and Physics, 2019_

## Referee Comment (RC1) · Anonymous Referee #1 · 25 Sep 2019

The authors examined the relationships between low-level cloud amount and the various proxies by low-level cloud types. This is worth studying and this study investigated the relationship extensively. In addition, the authors successfully showed advantages of their proxy ELF. I basically admit the scientific values of the results and the discussions. However, I have some concerns. It will be acceptable after concerns are addressed.

**Major Comments:**

1. Sizes of figures and characters in figures

Sizes of figures and characters in figures are too small to see. Figures 2, 3 and 4 should be much larger. I recommend the authors to move panels of $z_{LCL}$, $z_{inv}$, $\alpha$, and $1-\beta_2$ in these figures to supplement, and to divide Figs. 2 and 3 further in order to make the panels larger. Sizes of characters in Fig. 6, 7 should be larger. It is also desirable that sizes of tic marks of color bars in Fig. 1 and sizes of characters in Fig. 8 are larger.

2. Labels for cloud types

Cloud types are labeled as CL11, CL6, CL5, … I understand that they are labels based directly on the WMO classification and they have some advantages. However, it is very complicated when we read the manuscript because readers cannot easily remember the labels. Could you relabel them as, for instance, Fog, St, Sc, … or FOG, ST, SC, …, or CL_Fog, CL_St, CL_St, …?

3. Short physical explanations are needed in many parts

In many parts in the text, physical explanations that attribute the results to the characteristics of proxies are not enough. I guess they are helpful for readers even if they are just one or a few sentences. For example:

P6L22-23:

"both LTS and EIS increase, particularly over the far northern continents and Arctic area."
Please provide a suggestion of the reason why LTS and EIS increase in the situation.

P6L32-33:

"undesirable negative anomalies of LTS and EIS over the far northern continents including Arctic area get worse from CL11 to CL6 and CL7"

Please provide an interpretation of the reason why LTS and EIS show negative anomalies.

P7L5-7: "over the Arctic, Asia, and deserts areas, LTS/EIS shows negative anomalies opposite to the increased LCA, which worsens and extends to other continents from CL5, CL84 to CL12 and CL39"

Please provide a suggestion of the reason why LTS/EIS shows negative anomalies over the areas.

P7L22: "LTS and EIS, which have strong ocean-land contrasts (in particular, EIS) and seasonal cycle over land."

Please explain why ELF does not have strong ocean-land contrasts and seasonal cycle over land but LTS and EIS have them.

P7L24: "with a larger ELF during the night"

Please explain why ELF is larger during the night.

P7L34: "with systematically higher proxy values"

Can you guess why night slopes have systematically higher proxy values?

P7L34-35:

"both ELF and $1-\beta_2$ tend to have steeper regression slopes during the night than during the day"

Can you guess why regression slopes are steeper during the night than during the day?

Fig. 5c: The CL0 plots in Fig. 5c are against our simple tuition from previous studies (e.g., Wood and Bretherton (2006), Kawai et al. (2017)). This may confuse readers. Please briefly explain the reason of the apparent difference between CL0 plots in Fig. 5c and conventional figures.

P8L15: "The frequency of CL0 increases as LTS and EIS increase"

This is against our simple intuition, at least, over the ocean. What causes this increase over the ocean? Mainly where? In what season and what situation?

P8L32: "The freezedry factor substantially contributes to the improved correlations of CL0 with ELF from $\beta_2$"

Please briefly explain the physical meaning (for example, where and in what situation the factor mainly contributes to the improvement of the correlations).

P8L33-34:

"the frequent occurrence of CL0 on the west coast of the major continents and equatorial SST cold regions"

I guess that people do not expect that the occurrence of CL0 is frequent on the west coast of the major continents. Please add a little more explanation or note.

4. Target areas of LTS, EIS, and ECTEI

Please emphasize repeatedly in the text for fairness that the target areas of LTS, EIS, and ECTEI are over the ocean without sea ice and it is not intended to be used over land and sea ice.

5. Comparison of EIS and LTS

It is well-known that EIS is an index much better than LTS over the ocean. However, it is not so clear in the author's study. I guess readers will be confused. Please discuss a little why the superiority of EIS to LTS over the ocean is not clear in this study.

6. Discuss pros and cons of ELF compared with LTS/EIS/ECTEI.

Pros are very clear, I guess. Cons of ELF could be, for example:

* LTS/EIS/ECTEI tend to represent optically thick stratocumulus. It is important for earth radiation budget. Can ELF be directly used for discussions related to radiation budget?

* LTS/EIS/ECTEI are based on very simple concept. ELF and the proposed idea for improvement of ELF seem to be very empirical.

(* Discussion utilizing ELF or improved ELF could be complicated to understand LCA or LCA changes.)

(* LTS/EIS/ECTEI are very simple and easily calculated.)

7. Section 3.5

I'm afraid that proposed idea for improvement of ELF is too much empirical and complicated, although I understand the value of the challenge. Is it needed to construct a unified proxy for LCA by making a tremendous effort, even though the cloud regimes and mechanisms that produce LCA are quite different? Please discuss it a little.

8. Short discussion on cloud feedback

In the first paragraph of the introduction, the manuscript mentions an importance of the impact of low-level clouds on the Earth's climate including cloud feedback and climate sensitivity. However, there are no descriptions or suggestions on cloud feedback later in the manuscript, although this is a critically important topic now. Although the manuscript does not discuss it at all, proxies LTS, EIS, and ECTEI cause quite different estimation of cloud feedback. LTS causes strong negative cloud feedback, EIS suggests weak negative feedback, and ECTEI suggests positive cloud feedback over the ocean (models and observations imply positive cloud feedback, that is, a decrease in low-cloud in warmer climates). Could the authors add a short discussion or comments on cloud-feedback based on ELF?

**Minor comments:**

Somewhere:
  Is a variable $\beta_2$ defined somewhere?

P1L8-9: "the decrease in LCA when CL0 is reported and the increase of LCA when CL12 is reported"
  Are "decrease" and "increase" appropriate? It is not easy to understand, especially if readers don't read the contents yet, I guess.

P1L13: "the dissipation of LCA"
  Is the word "dissipation" appropriate?

P7L31: "a high EIS located outside of the plotting range"
  Can't you widen the range of the figure?

P8L5:  "Figure 6 is the cumulative plot"

Caption in Fig.6: "Cumulative FQ"

    Is Fig. 6 a cumulative plot? I though this is just a percentage plot.

Caption of Fig. 5:

    Explain the difference between open and filled symbols.

Fig. 6c, 6e:

    Why do LTS (and also EIS) have a large difference between daytime and night
    time over the ocean? It is understandable that there is a large difference over
    land (LTS and EIS is smaller in daytime). But why over the ocean also? I
    thought diurnal variations of LTS and EIS is negligible over the ocean because
    the SST diurnal variation is very small.

Fig.6e:  Please briefly explain the reason why the black line is very insensitive to EIS
    over the ocean. I guess many readers will be embarrassed because they often
    see the very clear relationship between LCA and EIS over the ocean in several
    papers (e.g., Wood and Bretherton (2006), Kawai et al. (2017)). Please clarify
    the cause of the differences.

Caption of Fig. 6:

    100 -> 100 %

---

## Referee Comment (RC2) · Anonymous Referee #3 · 24 Jan 2020

**Review of Shin & Park (2019), The Relationship Between Low-Level Cloud Amount and Its Proxies over the Globe by Cloud Types**

**Overview**

In Shin & Park (2019), the authors compare correlations between their estimated low-level cloud fraction (ELF) and other metrics like LTS and EIS for different low-level cloud types (as defined by a WMO-based typology).

ELF correlates better with cloud fraction than LTS or EIS for various different cloud types. Various components of ELF are also tested. All so-called "proxies" for cloud fraction struggle to diagnose cloud-free scenes. Suggestions are made to improve the ELF formulation to better account for cloud-free conditions and conditions in which cumulus updrafts occur in a background of more stratiform clouds and to parameterize the "scale height" as a function of environmental variables.

The analysis is extensive but not always well-conceptualized and the presentation needs significant work. In particular, it is not clear why it would be expected that LTS/EIS would correlate well with deep convective cloud types given their main use in analyzing shallow convective clouds or why we should expect to be able to approximate both shallow and deep convective cloud fraction with a single equation. Some variables that appear to be important and linked to the ELF derivation are left undefined. Figures are over-crowded and difficult to read. The overuse of jargon in terms of the cloud type abbreviations makes the reader's job very difficult.

Overall, substantial revisions are necessary to re-focus the manuscript, ideally to tell a more compelling story in the main narrative and perhaps provide more information as supplementary information (I rarely recommend moving material to a supplement, but here that may be quite helpful).

A suitably revised manuscript could be quite helpful for the modeling community and anyone else interested in estimating low cloud fraction and understanding its meteorological controls. I will happily recommend a revised version of this work be published in ACP assuming the authors are able to justify some of the choices made (or reduce the focus to what can be justified) and improve the presentation and organization of the manuscript.

**Major issues**

**1. Jargon**

The almost exclusive use of cloud type numbers (e.g., CL12) makes this paper extremely difficult to follow. (As a side note, "CL" is not a terribly intuitive abbreviation of cloud type either.) Table 1 is helpful but not sufficient, and does not list the combined types defined by the authors.

The authors should standardize how they describe each major cloud classification used (e.g., CL12 could be "shallow-to-moderate cumulus") and try to pair the descriptive words with the cloud type number as often as possible. Page 8, Line 29 does this very well — something like this should be done for the entire paper (including figure captions).

**2. Treatment of LTS, EIS, and ECTEI**

I am confused by the authors' treatment of LTS and EIS as low cloud "proxies" rather than as cloud-controlling factors. Clearly LTS and EIS correlate with stratiform clouds, but the strength of the boundary layer inversion is really only one relevant factor among several in explaining low cloud behavior. LTS/EIS can certainly be used as proxies for low cloud fraction, but this is not their primary/sole purpose.

Similarly, LTS/EIS really don't "diagnose" anything (e.g., Page 8, Lines 19-20). They are cloud-controlling factors (one of many!), not simple diagnostics in and of themselves.

This conceptual treatment leads to several statements that sound off, at least to my ears. For instance, on Page 9, Lines 4-5, is it truly "undesirable" that we can associate particularly large values of LTS/EIS with cloud clearing? This could be a useful observation to better understand potentially non-linear cloud behavior. This seems to me like a strange way to conceptualize LTS/EIS and why one would examine these variables.

The authors mention ECTEI in the abstract and (barely) define it in the introduction before noting it is similar to EIS and therefore not shown at the end of the Methods section. I would recommend having a supplement with the ECTEI results or not mentioning it at all (or only as a parenthetical). As written, the authors appear to promise an analysis they do not deliver.

**3. Definition of "low-level" cloud and its reasonableness**

While the observer-based methods define deep convection as "low-level" cloud based on the cloud base, there should be some discussion/reflection of whether this is a reasonable treatment in this analysis. LTS/EIS really are meant to explain cloud behavior in shallow boundary layers, not in deep convection. I don't particularly understand why we should expect one equation or metric to apply globally for both shallow and deep convection. If the authors do have a good explanation for this, it would be very helpful to provide it.

**4. Missing variable in the derivation of ELF**

Many times in the manuscript, the authors refer to and analyze a factor $(1 - \beta_2)$, but this is never defined. Please address this in the methods section. It also might be possible to reorganize the section deriving ELF to be more clear, especially with an eye toward the issues brought up in the final discussion of possible improvements for an "advanced ELF." Although the finer details of the ELF calculation addressed previously do not need to be explained in great detail, it should not be expected that all readers are familiar with PS19.

**5. General presentation and organization of figures**

The figures are far too crowded, and each subpanel much too small, to be easily interpreted by readers. In Figures 1-3, the black contours showing the climatology are nearly illegible. For Figure 1, a suggestion could be to split the figure up by cloud type (as is done for Figures 2-3) and have an added column for the climatology in its own map.

For Figures 2-3, I would also recommend subdividing further. One solution could be to have one figure include ELF and comparisons to LTS/EIS in one figure and the components of ELF in another. This could also help structure the discussion — first the differences between ELF, LTS, and EIS can be discussed, and then the contributions of the different components of ELF can be discussed.

It may also be a good idea to split up Figure 4 in a similar manner.

In Figure 5, the caption should explain that the color scheme is the same as that used in Figure 4. The open versus closed symbols also are not defined, although I assume they relate to day and night.

For the regressions in Figure 5, it would be good to address to what extent CL11 drives the regressions. Especially for subpanels b) and d), the scatter of points excluding CL11 (and CL0 and CLIM) do not appear to be very strongly correlated.

In Figure 8, the caption should make more clear that the adjustable scale height as a function of the environmental variables in g) and h) is shown as the "viridis" shading and is in units of meters.

**6. Interpretation of ELF correlation with cumulus cloud fraction in Tables 2 and 3**

On Page 12, Line 12, the authors write that ELF captures variations in cumulus clouds (CL12) better than LTS and EIS. Unless there is a typo in the tables, this is contradicted by the evidence provided in Tables 2 and 3. The global correlation of ELF with CL12 is ~0.03 whereas it is between -0.45 and -0.75 for LTS and EIS. Or is this sentence actually referring to CL84? In that case, the correlations are more all over the map. In any event, this is another good example of where the elimination of jargon in favor of clearly indicating which cloud type is being discussed would be helpful.

**Specific issues**

Page 1, Line 18: As the citation of Klein & Hartmann (1993) suggests, the efforts to quantify low cloud effects on Earth's climate long predate the last decade.

Page 2, Line 14: If you do choose to include ECTEI, its definition needs more exposition here.

Page 3, Eq. (5): It would be helpful to discuss that you then force the inversion height to lie between the LCL and the LCL plus a scale height in your analysis here. It's easy to miss as written. Also, for shallow convection, there's essentially no way for the inversion height to exceed the LCL plus scale height, right?

 Page 4, Line 9: "f" does not denote the amount of water vapor, it is a function of water vapor.

Page 4, Line 25: Individual components of ELF really aren't "proxies" for low cloud fraction by themselves. It would be more straightforward to just discuss these as components of ELF.

Page 4, Line 32: It would be helpful to explain that cloud types 12, 84, and 39 are actually combinations of types 1+2, 8+4, and 3+9.

Page 5, Lines 15-16: Moisture supply is not the only difference between marine and continental boundary layers (different responses to diurnal solar heating comes to mind as potentially being important here too).

Page 5, Line 25: I would expect the relative humidity to matter more than the total amount of moisture here, no?

Page 5, Lines 28-29: It would be helpful here to discuss how much of the advantage ELF has over LTS/EIS/ECTEI is due to the freezedry factor alone.

Page 6, Lines 5-7: Why isn't the composite analysis shown? It could at least be included in a supplement. The result isn't particularly surprising but would be interesting to see.

Page 6, Line 10: Why is there no hemispheric asymmetry in stratocumulus amount? If meteorology is the main driver, one would expect the hemispheric trends to be out of phase. In the Southern Hemisphere, the seasonal cycle tends to peak in spring and trough in fall whereas the Northern Hemisphere tends to peak in summer and trough in winter, so perhaps only looking at JJA-DJF differences doesn't capture the Southern Hemisphere seasonality well. Discussing SON and MAM seasonality (even if not shown, or just put in supplement) could be useful here.

Page 6, Line 20: It would be helpful to explain why the non-centered correlation is computed in some sections a centered correlation is computed in others, and whether this has any implication for the interpretation of your results.

Page 7, Lines 25-27: The latent cooling effect of evaporation should also matter for lowering the LCL.

Page 7, Line 31: Please either indicate what the outlier value is on the plot or report it here.

Page 8, Section 3.3: It would be helpful somewhere here to explain clearly what the difference between LCA and AMT is and how this should be interpreted.

Page 9, Line 28: "What is necessary" should replace "What are necessary".

Page 12, Line 24: What does the "(stratiform clouds FQ)" mean here in context? Is it supposed to refer to an increase in stratiform clouds as cumuliform cloud FQ decreases?

Page 13, Line 6: What would a negative depth for the decoupled layer mean physically? Wouldn't it just make more sense to define ELF piecewise rather than as a continuous function to account for these types of circumstances?

Page 13, Line 12: I do not understand what the "if any" means here. Surely you believe there is some appropriate variable, or why even discuss parameterizations of the scale height?

Page 13, Line 18: It would be good to list the download site for the ERA data here as well.

---

## Author Comment (AC1) · 10 Feb 2020

Thank you very much for the very valuable comments. We have addressed all your comments and revised the manuscript following your comments. We attached a zip file, which contains a response to the comments, the tracked-change version of the revised manuscript, and supplement of the revised manuscript. Sungsu

Please also note the supplement to this comment:
https://www.atmos-chem-phys-discuss.net/acp-2019-560/acp-2019-560-AC1-supplement.zip

---

## Author Response (AR1)

**Response to Anonymous Referee #1**

Thank you very much for your constructive and careful comments. It was greatly helpful to improve the quality of the draft.

**Note) Following the request of Anonymous Referee #3, Figure 1 was divided into Figure 1 and 2. So, the figure numbers of subsequent figures were increased by 1.

**Major Comments:**

**1 . Sizes of figures and characters in figures**

Sizes of figures and characters in figures are too small to see. Figures 2, 3 and 4 should be much larger. I recommend the authors to move panels of $z_{LCL}$, $z_{inv}$, $\alpha$, and $1-\beta_2$ in these figures to supplement, and to divide Figs. 2 and 3 further in order to make the panels larger. Sizes of characters in Fig. 6, 7 should be larger. It is also desirable that sizes of tic marks of color bars in Fig. 1 and sizes of characters in Fig. 8 are larger.

➡ Thank you for the suggestion. Following the comment, we divided Figures 3, 4, and 5. In addition, the panels of $z_{LCL}$, $z_{inv}$, $\alpha$, and $1-\beta_2$ are moved to supplement (S1, S2, S3). The size of the characters in Fig. 7, 8 is enlarged, and the sizes of tic marks of color bars in Fig. 2 and the sizes of characters in Fig. 9 are enlarged too.

**2 . Labels for cloud types**

Cloud types are labeled as CL11, CL6, CL5, ... I understand that they are labels based directly on the WMO classification and they have some advantages. However, it is very complicated when we read the manuscript because readers cannot easily remember the labels. Could you relabel them as, for instance, Fog, St, Sc, ... or FOG, ST, SC, ..., or CL_Fog, CL_St, CL_St, ...?

➡ Following the comment, we relabeled all the cloud types, and Table 2 is added to explain the abbreviations. Please see P5L10-13 in the tracked-change version.

**3 . Short physical explanations are needed in many parts**

In many parts in the text, physical explanations that attribute the results to the characteristics of proxies are not enough. I guess they are helpful for readers even if they are just one or a few sentences. For example:

P6L22-23:

"both LTS and EIS increase, particularly over the far northern continents and Arctic area."

Please provide a suggestion of the reason why LTS and EIS increase in the situation.

➔ This is because noCL (no low-level cloud) can occur when inversion is strong near the surface under dry conditions. We added the explanation in P7L10-12.

P6L32-33:

"undesirable negative anomalies of LTS and EIS over the far northern

continents including Arctic area get worse from CL11 to CL6 and CL7"

Please provide an interpretation of the reason why LTS and EIS show negative anomalies.

➔ We speculate that in these dry regions, the formation of Fog (CL11), F.St (CL6), and B.St (CL7) needs upward moisture transports from the surface, which is likely to be accompanied by the reduction of vertical stability in the lower troposphere. We added the explanation in P7L25-28.

P7L5-7: "over the Arctic, Asia, and deserts areas, LTS/EIS shows negative anomalies opposite to the increased LCA, which worsens and extends to other continents from CL5, CL84 to CL12 and CL39"

Please provide a suggestion of the reason why LTS/EIS shows negative anomalies over the areas.

➔ The negative correlation for Sc (CL5) can be explained by the same physical processes applied to the cases of Fog, F.St, and B.St as explained above. In the very dry regions where background LCA is very small, the onset of Cu (CL12) and Cb (CL39) in the low LTS/EIS

situations will result in the increase of LCA. We added the explanation in P8L3-6.

**P7L22: "LTS and EIS, which have strong ocean-land contrasts (in particular, EIS) and seasonal cycle over land."**

**Please explain why ELF does not have strong ocean-land contrasts and seasonal cycle over land but LTS and EIS have them.**

➔ The weaker seasonal cycle and ocean-land contrasts of ELF may imply the opposite variations in $z_{inv}$ and $z_{LCL}$. The freezedry factor also contributes to reducing the excessive seasonal cycle. We added the explanation in P8L21-22 and P8L23.

**P7L24: "with a larger ELF during the night"**

**Please explain why ELF is larger during the night.**

➔ This is presumably due in part to diagnosing of noCL condition as a non-zero ELF. We added the explanation in P8L24-25.

**P7L34: "with systematically higher proxy values"**

**Can you guess why night slopes have systematically higher proxy values?**

➔ It indicates that the product of $z_{inv}$ and $z_{LCL}$ during the day is larger than that during the night. We added the explanation in P9L1-5.

**P7L34-35:**

**"both ELF and 1-β2 tend to have steeper regression slopes during the night than during the day"**

**Can you guess why regression slopes are steeper during the night than during the day?**

➔ This is due in part to the diagnosis of noCL condition as a non-zero ELF, particularly, during

the night when noCL conditions are frequently reported. We added the explanation in P9L6-7.

**Fig. 5c: The CL0 plots in Fig. 5c are against our simple tuition from previous studies (e.g., Wood and Bretherton (2006), Kawai et al. (2017)). This may confuse readers. Please briefly explain the reason of the apparent difference between CL0 plots in Fig. 5c and conventional figures.**

➔ In responding to your comments above and below, we included explanations on this in P7L11-12 and P9L29-32 of the tracked-change version.

**P8L15: "The frequency of CL0 increases as LTS and EIS increase"**

**This is against our simple intuition, at least, over the ocean. What causes this increase over the ocean? Mainly where? In what season and what situation?**

➔ We note that noCL condition is frequently reported with a strong inversion at near the surface when LTS/EIS is large. We added the explanation in P9L29-32.

**P8L32: "The freezedry factor substantially contributes to the improved correlations of CL0 with ELF from β2"**

**Please briefly explain the physical meaning (for example, where and in what situation the factor mainly contributes to the improvement of the correlations).**

➔ As explained in PS19, the freezedry factor is designed to reduce a diagnosed LCA in a very dry region, such that it is most effective over the far northern continents and Arctic area, particularly during winter. We added the explanation in P10L16-18.

**P8L33-34:**

**"the frequent occurrence of CL0 on the west coast of the major continents and equatorial SST cold regions"**

**I guess that people do not expect that the occurrence of CL0 is frequent on the west coast of**

the major continents. Please add a little more explanation or note.

→ The frequent occurrence of noCL on the west coast is due to the advection of dry air from nearby continents. The frequent occurrence of noCL over the SST cold tongue is due to the warm air advection from the south. We added the explanation in P10L20-23

4 . Target areas of LTS, EIS, and ECTEI

Please emphasize repeatedly in the text for fairness that the target areas of LTS, EIS, and ECTEI are over the ocean without sea ice and it is not intended to be used over land and sea ice.

→ Following your comment, we emphasized repeatedly that the target area of LTS/EIS/ECTEI are over the ocean. Please see P7L29-30 and P11L16-17 in the tracked-change version.

5 . Comparison of EIS and LTS

It is well-known that EIS is an index much better than LTS over the ocean. However, it is not so clear in the author's study. I guess readers will be confused. Please discuss a little why the superiority of EIS to LTS over the ocean is not clear in this study.

→ It is well known that EIS is better than LTS in the marine stratocumulus deck regime. However, our analysis domain is not confined in the marine stratocumulus deck but extended into the entire globe with various cloud regimes. Because of this, it seems that the superiority of EIS over LTS is not clearly seen in our analysis. We briefly explained this in a revised draft in P9L11-14.

6 . Discuss pros and cons of ELF compared with LTS/EIS/ECTEI.

Pros are very clear, I guess. Cons of ELF could be, for example:

*   LTS/EIS/ECTEI tend to represent optically thick stratocumulus. It is important for earth radiation budget. Can ELF be directly used for discussions related to radiation budget?

*   LTS/EIS/ECTEI are based on very simple concept. ELF and the proposed idea for improvement of ELF seem to be very empirical.

(* Discussion utilizing ELF or improved ELF could be complicated to understand LCA or LCA

**changes.)**

**(\* LTS/EIS/ECTEI are very simple and easily calculated.)**

① **LTS/EIS/ECTEI tend to represent optically thick stratocumulus. It is important for earth radiation budget. Can ELF be directly used for discussions related to radiation budget?**

➔ ELF is designed to predict LCA of all types of clouds, so it can be used globally to discuss the radiation budget.

② **LTS/EIS/ECTEI are based on very simple concept. ELF and the proposed idea for improvement of ELF seem to be very empirical.**

➔ While the computation of ELF or improved ELF seems more complicated than LTS/EIS/ECTEI, we think that it is not so complicated. EIS needs $\theta_{sfc}$, $\theta_{700}$, $z_{LCL}$ and moist adiabatic lapse rates at $z_{LCL}$ and $z_{700}$ ($\Gamma_{LCL}$ and $\Gamma_{700}$) to calculate, and these are all information needed to calculate ELF too (if freezedry factor is ignored).

③ **Discussion utilizing ELF or improved ELF could be complicated to understand LCA or LCA changes**

➔ ELF (or improved ELF) can be useful to understand LCA changes. Please refer to the response to the comment below.

➔ It seems like the apparent con of ELF is that its formulation is bit complicated and empirical. We briefly discuss of pros and cons of ELF at P15L9-12 in the tracked-change version.

7. **Section 3.5**

**I'm afraid that proposed idea for improvement of ELF is too much empirical and complicated, although I understand the value of the challenge. Is it needed to construct a unified proxy for LCA by making a tremendous effort, even though the cloud regimes and mechanisms that produce LCA are quite different? Please discuss it a little.**

➔ The reason why we need a more precise unified proxy may be explained in relation to the cloud feedback. As shown in our paper, the response of LCA to environment variables is non-linear and varies depending on cloud types. Thus, to investigate the climate sensitivity of low-level clouds globally, it may be good to use a unified proxy, such as ELF. The contribution of individual environment variables can be extracted by linearizing ELF

formulation (e.g. $\Delta \text{ELF} \approx \frac{\partial \text{ELF}}{\partial z_{\text{inv}}} \Delta z_{inv} + \frac{\partial \text{ELF}}{\partial z_{\text{LCL}}} \Delta z_{LCL} + \frac{\partial \text{ELF}}{\partial f} \Delta f$). In this way, we can describe the physical processes controlling low cloud feedback, which depends on cloud regimes, in a single framework. As noted by the reviewer, the development of an advanced ELF may take lots of time and effort. However, due to the reasons mentioned above, we think it is worthwhile to do that. We briefly included this discussion in P15L9-12.

**8 . Short discussion on cloud feedback**

**In the first paragraph of the introduction, the manuscript mentions an importance of the impact of low-level clouds on the Earth's climate including cloud feedback and climate sensitivity. However, there are no descriptions or suggestions on cloud feedback later in the manuscript, although this is a critically important topic now. Although the manuscript does not discuss it at all, proxies LTS, EIS, and ECTEI cause quite different estimation of cloud feedback. LTS causes strong negative cloud feedback, EIS suggests weak negative feedback, and ECTEI suggests positive cloud feedback over the ocean (models and observations imply positive cloud feedback, that is, a decrease in low-cloud in warmer climates). Could the authors add a short discussion or comments on cloud-feedback based on ELF?**

➔ Thank you very much for the very nice comments. As noted by the reviewer, exploring cloud feedback and climate sensitivity is an extremely important subject. Following the comments, we examined the climate sensitivity diagnosed by ECTEI, LCA, and ELF. The below figure shows the SST dependency of ECTEI, LCA, and ELF over the ocean. ECTEI is one of the unified LCA proxies which accounts SST dependency of LCA by including cloud top entrainment criteria. As shown in the figure, ECTEI is tightly dependent on SST and EIS, but the scatters of LCA and ELF are more divergent. This implies that cloud controlling factors other than SST and EIS should work for the observed LCA. Both ECTEI and ELF predict the negative LCA slope to SST for a fixed EIS, which is known to compensate the LCA increase in a warm climate in association with higher EIS. The ELF-predicted SST slope is -0.66 % K$^{-1}$, which is smaller than that of LCA (-1.66) and ECTEI (-0.80). This result indicates a need to develop a more advanced ELF.

➔ As noted by the reviewer, exploring climate sensitivity is extremely important and a huge research subject. However, a detailed examination of climate sensitivity seems to be out of the main theme of our current draft, which focused on the relationship between LCA and various proxies by cloud types.

➔ So, we think that it will be better to explore climate sensitivity in a separate paper in a more comprehensive way, which, in fact, is one of our future research subjects. Following the comments, this future research plan is briefly explained in the conclusion section (P15L15-

17).

[Figure]

**Figure 1.** Scatter diagrams showing SST dependency of ECTEI, LCA, and ELF for each EIS bin (denoted by different colors). All seasonal climatologies of 5° latitude x 10° longitude ocean grids boxes between 60°N and 60°S are used in the analysis. The mean SST slope ($[\partial LCA/\partial SST]_{EIS}$ in units of $[\% \, K^{-1}]$) denoted at the top of each panel is calculated by doing a linear regression for each EIS bin and averaging the regression coefficients of all EIS bin. For ECTEI, a conversion factor between LCA and ECTEI is assumed as dLCA/dECTEI = 3.1 % $K^{-1}$ following Kawai et al. (2017).

**Minor comments:**

**Somewhere:**

**Is a variable β2 defined somewhere?**

➔ We added the definition of β2 in P3L21-23.

**P1L8-9: "the decrease in LCA when CL0 is reported and the increase of LCA when CL12 is reported"**

Are "decrease" and "increase" appropriate? It is not easy to understand, especially if readers don't read the contents yet, I guess.

➔ Following the comment, we removed the wording of "increase" and "decrease", and changed

them to "changes". Please see P1L8-9 in the tracked-change version.

**P1L13: "the dissipation of LCA"**

**Is the word "dissipation" appropriate?**

➔ The phrase "dissipation of" has been modified to "decrease of" for clarity. Please see P1L13 in the tracked-change version.

**P7L31: "a high EIS located outside of the plotting range" Can't you widen the range of the figure?**

➔ Following the comment, we have widened the range of the figure. Following the comment of referee #2, the squared regression coefficients ($R^2$) without Fog are added in Figure 6. Thus, we rewrote the sentence. Please see Figure 6 and P8L31-34 in the tracked-change version.

**P8L5:   "Figure 6 is the cumulative plot"**

**Caption in Fig.6: "Cumulative FQ"**

**Is Fig. 6 a cumulative plot? I though this is just a percentage plot.**

➔ The more appropriate name of the plot is "stacked percentage plot". Please see the caption of Figure7 and P9L16 in the tracked-change version.

**Caption of Fig. 5:**

**Explain the difference between open and filled symbols.**

➔ Thank you for pointing out. We added the explanation in the caption of Figure 6.

**Fig. 6c, 6e:**

**Why do LTS (and also EIS) have a large difference between daytime and night time over the ocean? It is understandable that there is a large difference over land (LTS and EIS is smaller in daytime). But why over the ocean also? I thought diurnal variations of LTS and EIS is negligible over the ocean because the SST diurnal variation is very small.**

➔ It seems that the reviewer mis-interpreted Fig.7c and 7e. As was explained in the caption of Fig.7, "The bright and dark colors in each bar denote the fractions during the daytime and nighttime, respectively", instead of representing the values of LTS (or EIS) during the daytime and nighttime.

**Fig.6e: Please briefly explain the reason why the black line is very insensitive to EIS over the ocean. I guess many readers will be embarrassed because they often see the very clear relationship between LCA and EIS over the ocean in several papers (e.g., Wood and Bretherton (2006), Kawai et al. (2017)). Please clarify the cause of the differences.**

➔ As explained before, the high correlation between EIS and LCA reported in previous studies is mainly for the case of stratocumulus (CL5, CL6, CL84). In our study, however, we are examining the correlation across the entire low cloud types, such that the correlation between EIS and LCA is not large, as shown in Fig.6. This explanation is added in P9L32-34.

**Caption of Fig. 6:**

   **100 -> 100 %**

➔ Corrected. Please see the caption of Figure 7.

**Response to Anonymous Referee #3**

Thank you very much for your constructive and careful comments. It was greatly helpful to improve the quality of the draft.

**Note) Following the request of Anonymous Referee #3, Figure 1 was divided into Figure 1 and 2. So, the figure numbers of subsequent figures were increased by 1.**

**Major issues**

**1. Jargon**

The almost exclusive use of cloud type numbers (e.g., CL12) makes this paper extremely difficult to follow. (As a side note, "CL" is not a terribly intuitive abbreviation of cloud type either.) Table 1 is helpful but not sufficient, and does not list the combined types defined by the authors.

The authors should standardize how they describe each major cloud classification used (e.g., CL12 could be "shallow-to-moderate cumulus") and try to pair the descriptive words with the cloud type number as often as possible. Page 8, Line 29 does this very well — something like this should be done for the entire paper (including figure captions).

➔ Following the comment, we relabeled all the cloud types, as explained in Table 2. Please see P5L10-12 in the tracked-change version.

**2. Treatment of LTS, EIS, and ECTEI**

I am confused by the authors' treatment of LTS and EIS as low cloud "proxies" rather than as cloud-controlling factors. Clearly LTS and EIS correlate with stratiform clouds, but the strength of the boundary layer inversion is really only one relevant factor among several in explaining low cloud behavior. LTS/EIS can certainly be used as proxies for low cloud fraction, but this is not their primary/sole purpose.

Similarly, LTS/EIS really don't "diagnose" anything (e.g., Page 8, Lines 19-20). They are cloud-controlling factors (one of many!), not simple diagnostics in and of themselves.

This conceptual treatment leads to several statements that sound off, at least to my ears. For instance, on Page 9, Lines 4-5, is it truly "undesirable" that we can associate particularly large values of LTS/EIS with cloud clearing? This could be a useful observation to better understand

potentially non-linear cloud behavior. This seems to me like a strange way to conceptualize LTS/EIS and why one would examine these variables.

The authors mention ECTEI in the abstract and (barely) define it in the introduction before noting it is similar to EIS and therefore not shown at the end of the Methods section. I would recommend having a supplement with the ECTEI results or not mentioning it at all (or only as a parenthetical). As written, the authors appear to promise an analysis they do not deliver.

➔ We used the term "proxy" for the LTS and EIS, in order to keep consistency with our previous paper (Park and Shin 2019; PS19) which already used LTS and EIS as one of LCA proxies. At least in stratiform cloud regions, LTS and EIS have been used as proxies of LCA in many papers. Many readers will be familiar with this and there won't be much difficulty in understanding the concept. However, we agree that several statements could confuse some readers. Thus, following the comment, we modified the following.

① "Clearly LTS and EIS correlate with stratiform clouds, but the strength of the boundary layer inversion is really only one relevant factor among several in explaining low cloud behavior."
➔ We agree with the comment and included this explanation in P2L14-15.

② "Similarly, LTS/EIS really don't "diagnose" anything (e.g., Page 8, Lines 19-20). They are cloud-controlling factors (one of many!), not simple diagnostics in and of themselves."
➔ Following the comment, we rephrased this sentence. Please see P10L1-3.

③ "on Page 9, Lines 4-5, is it truly "undesirable" that we can associate particularly large values of LTS/EIS with cloud clearing? This could be a useful observation to better understand potentially non-linear cloud behavior. This seems to me like a strange way to conceptualize LTS/EIS and why one would examine these variables. "
➔ We agree that the word "undesirable" is not appropriate here. Thus, we changed the word "undesirable" to "unexpected".
➔ We also noted that the strong positive correlation between LTS/EIS and noCL FQ might indicate a non-linear response of clouds to the inversion strength or the existence of other factors controlling noCL. Please see P10L27-30 in the tracked change version.

➔ In addition, we stated that the target areas of LTS, EIS, and ECTEI are over the ocean. Please see P7L29-30 and P11L16-17 in the tracked-change version.

③ **The authors mention ECTEI in the abstract and (barely) define it in the introduction before noting it is similar to EIS and therefore not shown at the end of the Methods section. I would recommend having a supplement with the ECTEI results or not mentioning it at all (or only as a parenthetical). As written, the authors appear to promise an analysis they do not deliver.**

➔ Following the comment, we removed ECTEI from the abstract. Although not shown, the analysis results of ECTEI are **almost identical** to EIS as mentioned at the end of the Methods section (P5L25). Thus, we did not include the results of ECTEI in the supplement.

3.      Definition of "low-level" cloud and its reasonableness

While the observer-based methods define deep convection as "low-level" cloud based on the cloud base, there should be some discussion/reflection of whether this is a reasonable treatment in this analysis. LTS/EIS really are meant to explain cloud behavior in shallow boundary layers, not in deep convection. I don't particularly understand why we should expect one equation or metric to apply globally for both shallow and deep convection. If the authors do have a good explanation for this, it would be very helpful to provide it.

➔ Because deep convection is controlled by similar physical processes as shallow convection (Park 2014a,b), it is unnecessary to use separate formulation for shallow and deep convections. In addition, at least in terms of cloud fraction, we thought that a decoupling hypothesis can describe the changes in cloud fraction from the well mixed (Sc), partially decoupled (Sc-Cu), and fully decoupled (Cu, Cb) conditions. This is the philosophy of ELF. We briefly included this explanation in P5L13-14.

4.      Missing variable in the derivation of ELF

Many times in the manuscript, the authors refer to and analyze a factor $(1 - \beta_2)$, but this is never defined. Please address this in the methods section. It also might be possible to reorganize the section deriving ELF to be more clear, especially with an eye toward the issues brought up in the final discussion of possible improvements for an "advanced ELF." Although the finer details of the ELF calculation addressed previously do not need to be explained in great detail, it should not be expected that all readers are familiar with PS19.

➔ Following the comment, the definition of (1 - β2) is added in P3L22-23 in the tracked-change version. We also reorganized the structure of explaining the definition of ELF (P3L21-P4L6). We did not add very detailed derivation of ELF here, because it requires a lengthy explanation of the conceptual framework with a diagram.

**5. General presentation and organization of figures**

**The figures are far too crowded, and each subpanel much too small, to be easily interpreted by readers. In Figures 1-3, the black contours showing the climatology are nearly illegible. For Figure 1, a suggestion could be to split the figure up by cloud type (as is done for Figures 2-3) and have an added column for the climatology in its own map.**

➔ Following the comment, we divided Figure 1 to Figure 1 and Figure 2.

**For Figures 2-3, I would also recommend subdividing further. One solution could be to have one figure include ELF and comparisons to LTS/EIS in one figure and the components of ELF in another. This could also help structure the discussion — first the differences between ELF, LTS, and EIS can be discussed, and then the contributions of the different components of ELF can be discussed.**
**It may also be a good idea to split up Figure 4 in a similar manner.**

➔ Following the comment, we divided Figures 3, 4, and 5 (previously Figures 2-4) and panels of $z_{LCL}$, $z_{inv}$, $\alpha$, and $1-\beta_2$ are moved to supplement (S1, S2, S3).

**In Figure 5, the caption should explain that the color scheme is the same as that used in Figure 4. The open versus closed symbols also are not defined, although I assume they relate to day and night.**
**For the regressions in Figure 5, it would be good to address to what extent CL11 drives the regressions. Especially for subpanels b) and d), the scatter of points excluding CL11 (and CL0 and CLIM) do not appear to be very strongly correlated.**

➔ In the caption of Figure 6, we explained that the color scheme used is the same as that used in Figure 5. The open and closed symbols are explained too.
➔ Following the comment, we also added squared regression coefficients ($R^2$) without Fog (CL11) in parenthesis. A corresponding explanation is written in P8L31-34 in the tracked-change version.

**In Figure 8, the caption should make more clear that the adjustable scale height as a function of the environmental variables in g) and h) is shown as the "viridis" shading and is in units of meters.**

➔ In the caption of figure 9, we specified that the adjustable scale height is shown as shading and in units of meters.

**6. Interpretation of ELF correlation with cumulus cloud fraction in Tables 2 and 3**

**On Page 12, Line 12, the authors write that ELF captures variations in cumulus clouds (CL12) better than LTS and EIS. Unless there is a typo in the tables, this is contradicted by the evidence provided in Tables 2 and 3. The global correlation of ELF with CL12 is ~0.03 whereas it is between -0.45 and -0.75 for LTS and EIS. Or is this sentence actually referring to CL84? In that case, the correlations are more all over the map. In any event, this is another good example of where the elimination of jargon in favor of clearly indicating which cloud type is being discussed would be helpful.**

➔ It seems that the reviewer misunderstood. Tables 3 and 4 do not show the correlations between proxies and LCA; they show the correlations between proxies and **the frequency (FQ)** of individual cloud type. If any proxy is perfect, the correlation between the perfect proxy and CL FQ should be identical to the correlation between the LCA and CL FQ.

➔ As an example: The global correlation between cumuli's LCA and FQ is 0.10. ELF has a similar correlation of -0.03. LTS and EIS have the correlation values of -0.45 and -0.75. In this case, ELF is a better proxy for LCA than LTS and EIS.

**Specific issues**

**Page 1, Line 18: As the citation of Klein & Hartmann (1993) suggests, the efforts to quantify low cloud effects on Earth's climate long predate the last decade.**

➔ We changed "last decade" to "past few decades". Please see P1L17 in the tracked-change version.

**Page 2, Line 14: If you do choose to include ECTEI, its definition needs more exposition here.**

➔ Following the comment, we added the definition of ECTEI in the Method section. Please see lines P3L17-20 in the tracked-change version.

**Page 3, Eq. (5): It would be helpful to discuss that you then force the inversion height to lie between the LCL and the LCL plus a scale height in your analysis here. It's easy to miss as written. Also, for shallow convection, there's essentially no way for the inversion height to exceed the LCL plus scale height, right?**

➔ Following the comment, the range of the inversion height is added in Eq. (6). Please see P3L25 in the tracked-change version. As you said, the inversion height cannot exceed LCL plus scale height, but since scale height is $\Delta z_s$ = 2750m, the upper limit of inversion height can easily exceed the height of 700hPa.

**Page 4, Line 9: "f" does not denote the amount of water vapor, it is a function of water vapor.**

➔ We specified that "f" is an increasing function of water vapor. Please see P4L14 in the tracked-change version.

**Page 4, Line 25: Individual components of ELF really aren't "proxies" for low cloud fraction by themselves. It would be more straightforward to just discuss these as components of ELF.**

➔ We rewrote the sentence. Please see P5L1-3 in the tracked-change version.

**Page 4, Line 32: It would be helpful to explain that cloud types 12, 84, and 39 are actually combinations of types 1+2, 8+4, and 3+9.**

➔ The combination of the cloud types are explained in Table2. Please see P5L10-13 in the tracked-change version.

**Page 5, Lines 15-16: Moisture supply is not the only difference between marine and continental boundary layers (different responses to diurnal solar heating comes to mind as potentially being important here too).**

➔ We specified that the moisture supply is "one of the important factors", rather than "primary factor". Please see P5L29-30 in the tracked-change version.

**Page 5, Line 25: I would expect the relative humidity to matter more than the total amount of moisture here, no?**

➔ In the far northern continents and Arctic area, the freezedry factor, which is a function of the absolute moisture amount, becomes very important for the onset of noCL. The relative humidity is also important but the amount of moisture is a more comprehensive concept.

**Page 5, Lines 28-29: It would be helpful here to discuss how much of the advantage ELF has over LTS/EIS/ECTEI is due to the freezedry factor alone.**

➔ First, we briefly explained why ELF is improved by the freezedry factor in P6L11-12. The quantitative improvements are already investigated in our previous study, so we cited the paper (PS19) here.
➔ The effect of the freezedry factor is discussed many times in subsequent sections (e.g. P7L13, P8L23).

**Page 6, Lines 5-7: Why isn't the composite analysis shown? It could at least be included in a supplement. The result isn't particularly surprising but would be interesting to see.**

➔ The composite is not shown here because it will be included in the paper we are preparing. We cited the paper so future readers could find corresponding figures. Please see P6L23 in the tracked-change version.

**Page 6, Line 10: Why is there no hemispheric asymmetry in stratocumulus amount? If meteorology is the main driver, one would expect the hemispheric trends to be out of phase. In the Southern Hemisphere, the seasonal cycle tends to peak in spring and trough in fall whereas the Northern Hemisphere tends to peak in summer and trough in winter, so perhaps only looking at JJA-DJF differences doesn't capture the Southern Hemisphere seasonality well. Discussing SON and MAM seasonality (even if not shown, or just put in supplement) could be useful here.**

➔ As Klein and Hartmann (1993) shown, stratiform clouds in the Namibian and Peruvian stratocumulus decks tend to peak in SON. Since the detailed analysis on the seasonal cycle is not the scope of our paper, we just cited Klein and Hartmann (1993) here. Please see P6L28-30 in the tracked change version.

**Page 6, Line 20: It would be helpful to explain why the non-centered correlation is computed in some sections a centered correlation is computed in others, and whether this has any implication for the interpretation of your results.**

→ We explained why the non-centered correlation is computed here. Please see P7L7-9 in the tracked change version.

**Page 7, Lines 25-27: The latent cooling effect of evaporation should also matter for lowering the LCL.**

→ Corrected. Please see P8L28 in the tracked-change version.

**Page 7, Line 31: Please either indicate what the outlier value is on the plot or report it here.**

→ We extended the range of x-axis of Figure 6, so the scatter located outside of the plot is now located inside of the plot. Please see Figure 6, and also see P8L31-34 in the tracked-change version.

**Page 8, Section 3.3: It would be helpful somewhere here to explain clearly what the difference between LCA and AMT is and how this should be interpreted.**

→ Following the comment, we added an explanation of the difference between LCA and AMT in Section 3.4. Please see P10L32-33 in the tracked-change version.

**Page 9, Line 28: "What is necessary" should replace "What are necessary".**

→ Corrected. Please see P11L20 in the tracked-change version.

**Page 12, Line 24: What does the "(stratiform clouds FQ)" mean here in context? Is it supposed to refer to an increase in stratiform clouds as cumuliform cloud FQ decreases?**

→ "Increase in" is mistakenly omitted here, so we corrected it. Please see P14L19 in the tracked-change version.

**Page 13, Line 6: What would a negative depth for the decoupled layer mean physically? Wouldn't it just make more sense to define ELF piecewise rather than as a continuous function to account for these types of circumstances?**

➔ Following the comment, we explained the physical meaning of a negative decoupled layer depth at P12L1-2 in the tracked-change version.

➔ As you commented, it can be one option to define ELF piecewise by separating the cases where a decoupled layer has negative depth or positive depth. However, such a strategy does not seem to work well when we tested it. Probably because the calculation of the inversion height is not accurate.

**Page 13, Line 12: I do not understand what the "if any" means here. Surely you believe there is some appropriate variable, or why even discuss parameterizations of the scale height?**

➔ It seems like "if any" is unnecessary here, so it is deleted. Please see P15L7 in the tracked-change version.

**Page 13, Line 18: It would be good to list the download site for the ERA data here as well.**

➔ Following the comment, we listed the download site for the ERA data. Please see P15L18-19 in the tracked-change version.

[revised manuscript text omitted]